# BEEF: BUILDING A BRIDGE FROM EVENT TO FRAME

## ABSTRACT

Event-based cameras are attracting significant interest as they provide event streams that contain rich edge information with high dynamic range and high temporal resolution. Many state-of-the-art event-based algorithms rely on splitting the events into several fixed groups, which are then aggregated into 2D frames by different event representations. However, the fixed slicing method can result in the omission of crucial temporal information, particularly when dealing with diverse motion scenarios (*e.g.*, high-speed and low-speed). In this work, to build a **B**ridg**E** from converting **E**vent streams to **F**rames, we propose **BEEF**, a novel-designed event processing framework capable of splitting events stream to frames in an adaptive manner. In particular, BEEF integrates a low-energy spiking neural network (SNN) as an event trigger to determine the slicing time based on the spike generation. To guide the SNN in firing spikes at optimal time steps, we introduce the Spiking Position-aware Loss (SPA-Loss) to modulate the neuron's spiking state. In addition, we develop a novel Feedback-Update training strategy that supervises the SNN to make precise event slicing decisions based on the feedback from the downstream artificial neural network (ANN). The newly sliced dataset by SNN is then used to finetune the ANN to improve the overall performance. Extensive experiments demonstrate that our BEEF achieves state-of-the-art performance in event-based object tracking and recognition. Notably, BEEF provides a brand-new SNN-ANN cooperation paradigm, where the SNN acts as an efficient, low-energy data processor to assist the ANN in improving downstream performance, injecting new perspectives and potential avenues of exploration.

## 1 INTRODUCTION

Event-based cameras (Gallego et al., 2020) are bio-inspired sensors that capture event streams in an asynchronous and sparse way. Compared with conventional frame-based cameras, event-based cameras offer numerous outstanding properties: high temporal resolution (with the order of $\mu s$), high dynamic range (higher than 120 dB), low latency, and low power consumption. Over recent years, rapid growth has been witnessed in dealing with event data due to the inherent advantages of event-based cameras, such as object tracking (Zhang et al., 2022a; 2021) and detection (Cao et al., 2023b), depth estimation (Zhang et al., 2022b; Nam et al., 2022), and recognition (Sironi et al., 2018; Baldwin et al., 2022). Events are usually processed by using frame-based frameworks due to a large number of state-of-the-art deep learning models available (Fang et al., 2021; Zheng et al., 2021). Before applying to various downstream tasks, the event stream must be split by groups and then transformed into a dense representation for frame-based architectures.

In details, the process of event-to-frame conversion is mainly divided into two steps: **(1) slicing the raw event stream into multiple sub-event stream groups**, and **(2) convert these sub-event streams into frames using various event representation methods.** Much of the current research focuses on the second step, aiming to refine event representation (Wang et al., 2019; Sironi et al., 2018) techniques such as time surface and event spike tensor (Gehrig et al., 2019). Yet, this focus often overlooks the crucial first step of slicing, where issues such as non-uniformity in event distribution in varying motion speeds remain unaddressed.

To address the gap happened in the slicing process, we delve into the limitations of traditional slicing techniques. Common methods typically cut the event stream into several fixed groups. For example, slicing event stream with fixed event count (Maqueda et al., 2018) or fixed time intervals (Zhu et al., 2022; 2019). However, these fixed-group slicing techniques often lead to problems: they may

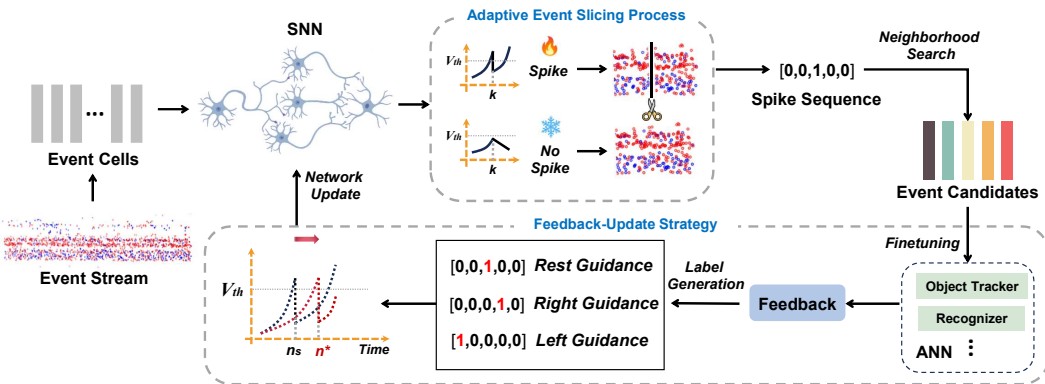

Figure 1: Overview of our BEEF framework. The input events are first passed through an SNN, and the event is determined to be sliced when a spike occurs. To find the accurate slicing time, the neighborhood search method explores other time steps and feeds event candidates to the downstream ANN model (*e.g.*, object tracker or recognizer). The ANN model then offers feedback, which guides the SNN in firing spikes at the optimal slicing time by supervising the membrane potential. Subsequently, the updated SNN generates a new event dataset and fine-tunes the ANN parameters. This process is iterated to improve the overall performance.

result in insufficient information capture in high-speed motion scenarios or excessive redundancy in low-speed conditions, failing to accurately capture the dynamic changes in event distribution. Additionally, some hyper-parameters, *e.g.*, the length of time interval, are highly-sensitive to the event-to-frame process (proved in Appendix B) and must be carefully pre-determined. Although some latest slicing method (Peng et al., 2023; Li et al., 2022) proposes to adaptively sampling the events, there still exists the problem of hyper-parameter tuning which can not achieve a fully learnable and adaptable slicing process.

In order to address the above issues, we propose BEEF, a novel-design event processing framework that can slice the event streams in an adaptive manner. To achieve this, BEEF utilizes an SNN as an event trigger to dynamically determine the optimal moment to split the event stream. Our objectives include: (1) training the SNN to spike at a specific time step, and (2) developing a training strategy to identify the best slicing time for a continuous event stream during training. In our paper, we achieve (1) through our newly introduced Spiking Position-aware Loss (SPA-Loss) function, which effectively guides the SNN to spike at the desired time by manipulating the membrane potential. For (2), we implement a Feedback-Update training strategy, where the SNN receives real-time performance feedback from the downstream ANN model for supervision. An overview of our proposed framework is depicted in Fig. 1. We evaluate the effectiveness of our proposed BEEF in two downstream tasks: (i) event-based object tracking, which is strongly sensitive to temporal information and motion dynamics, and (ii) event-based recognition, which is highly related to event density. Extensive experiments validate the effectiveness of the proposed approach.

To sum up, our contributions are as follows:

- We propose BEEF, a novel-designed event processing framework capable of splitting event streams for generating event frames in an adaptive manner.

- We design the SPA-Loss to guide the SNN to trigger spikes at the expected time steps. We then propose a novel Feedback-Update strategy that optimizes the event slicing process based on the ANN feedback.

- Extensive experiments demonstrate that BEEF significantly improves the model performance in event-based tracking and recognition with high processing speed (111 FPS on GPU) and ultra-low energy consumption ($0.022mJ$ per image). Notably, BEEF provides a brand-new SNN-ANN cooperation paradigm, where the SNN serves as an efficient, low-energy data processor to assist the ANN in improving downstream performance.

## 2 BACKGROUND AND RELATED WORK

**Event-based Cameras.** They are bio-inspired sensors, which capture the relative intensity changes asynchronously. In contrast to standard cameras that output 2D images, event cameras output sparse event streams. When brightness change exceeds a threshold $C$, an event $e_k$ is generated containing position $\mathbf{u} = (x, y)$, time $t_k$, and polarity $p_k$:

$$\Delta L(\mathbf{u}, t_k) = L(\mathbf{u}, t_k) - L(\mathbf{u}, t_k - \Delta t_k) = p_k C. \tag{1}$$

The polarity of an event reflects the direction of the changes (*i.e.*, brightness increase ("ON") or decrease ("OFF")). In general, the output of an event camera is a sequence of events, which can be described as: $\mathcal{E} = \{e_k\}_{k=1}^N = \{[\mathbf{u}_k, t_k, p_k]\}_{k=1}^N$. With the advantages of high temporal resolution, high dynamic range, and low energy consumption, event cameras are gradually attracting attention in the fields of tracking (Zhang et al., 2021; 2023), identification (Baldwin et al., 2022) and estimation (Nam et al., 2022).

**Spiking Neural Network (SNN).** SNNs are potential competitors to artificial neural networks (ANNs) due to their distinguished properties: high biological plausibility, event-driven nature, and low power consumption. In SNNs, all information is represented by binary time series data rather than float representation, leading to significant energy efficiency gains. Also, SNNs possess powerful abilities to extract spatial-temporal features for various tasks, including recognition (Wang et al., 2022; Zhou et al., 2022), tracking (Zhang et al., 2022a), detection (Kim et al., 2020) and image generation (Cao et al., 2023a). In this paper, we adopt the widely used SNN model based on the Leaky Integrate-and-Fire (LIF, (Hunsberger & Eliasmith, 2015; Burkitt, 2006)) neuron, which effectively characterizes the dynamic process of spike generation and can be defined as:

$$V[n] = \beta V[n-1] + \gamma I[n], \tag{2}$$
$$S[n] = \Theta(V[n] - \vartheta_{\text{th}}), \tag{3}$$

where $n$ is the time step and $\beta$ is the leaky factor that controls the information reserved from the previous time step; $V[n]$ is the membrane potential; $S[n]$ denotes the output spike which equals 1 when there is a spike and 0 otherwise; $\Theta(x)$ is the Heaviside function. When the membrane potential exceeds the threshold $\vartheta_{\text{th}}$, the neuron will trigger a spike and resets its membrane potential to $V_{\text{reset}} < \vartheta_{\text{th}}$. Meanwhile, when $\beta = \gamma = 1$, LIF neuron evolves into Integrate-and-Fire (IF) neuron. We also introduce a no-reset membrane potential $U[n]$, meaning that the membrane potential does not reset, but directly passes the original value to the next time step (*i.e.* $U[n] = V[n]$ after Eq. 3).

## 3 MOTIVATION

Event cameras, which are bio-inspired sensors that exhibit remarkable advantages over RGB cameras, have recently drawn great attention since they provide rich edge information with high dynamic range (HDR) and high temporal resolution. However, existing event-based algorithms face several challenges:

**Challenge 1: Slicing the event stream in a fixed form neglects crucial spatio-temporal information of events.** Existing event-to-frame algorithms require slicing the event stream into $N$ grids by fixed method. However, due to the high temporal resolution and non-uniformity of event streams, the information within each slice may be less informative or redundant. Meanwhile, the performance of downstream tasks is highly sensitive and unstable to the number of $N$, as proven by Appendix B.

**Challenge 2: Existing event-to-frame methods are completely unrelated to the downstream tasks.** The current event-to-frame method lacks a connection to downstream tasks, which constrains the ability of downstream models to comprehend the raw data effectively.

In summary, converting event streams into frames remains intractable. Based on the aforementioned analysis, we outline the motivations of our proposed approach:

---
**Motivations:**

1. Fixed-sliced event representation obscures crucial timing information.
— Dynamically slicing the event stream is an intuitive solution.

2. There is no connection between event-to-frame methods and downstream models.
— Establish a "downstream update, upstream feedback" strategy.

---

# 4 OUR APPROACH: BEEF

In this section, we first introduce the process of event stream conversion to properly obtain the event representation by introducing the concept of event cells (Sec. 4.1). Then, we introduce the adaptive event slicing process by utilizing an SNN as the event trigger (Sec. 4.2). In Sec. 4.3, we introduce a novel Spiking Position-aware Loss (SPA-Loss) to supervise the SNN to slice the event at the precise time. Finally, we build a feedback-update (Sec. 4.4) strategy that allows the resulting events to be correlated with the feedback from the downstream model, thereby improving overall performance.

## 4.1 CONVERTING EVENT STREAM TO EVENT CELL

Event streams are asynchronous data that can be represented as a set: $\mathcal{E} = \{[x_i, y_i, t_i, p_i]\}_{i=1}^{N}$ with a time span of $T$ (*i.e.*, $t_i \in [t_0, t_0 + T]$). However, in software simulations, the event stream should be converted into dense grid-like representations to comply with the input requirements of existing deep learning frameworks. This means we must find a mapping $\mathcal{M} : \mathcal{E} \mapsto \mathcal{T}$ between the set $\mathcal{E}$ and a tensor $\mathcal{T}$. Therefore, we first introduce the event cell:

**Definition 1** (Event cell). *Consider a small time interval $\delta t$, event cell is a single-grid event representation in the form of: $C_{\pm}(x, y, t_*) = \mathcal{F}(G_{\pm}(x, y, t, \{t \in [t_*, t_* + \delta t]\}))$, where $\mathcal{F}$ denotes an aggregated method to stack the event representation $G_{\pm}$ (Appendix E) with $t \in [t_*, t_* + \delta t]$ into a 2D representation.*

A whole event stream can be then represented by a list of $N$ event cells, *i.e.*, $\{C_{\pm}(x, y, t_0), C_{\pm}(x, y, t_0 + \delta t), ..., C_{\pm}(x, y, t_0 + (N-1)\delta t)\}$, where $N = T/\delta t$ and each cell corresponds to a discrete time index $n \in \{0, 1, ..., N-1\}$. In the following sections, we abbreviate the event cell by $C[n]$ for simplicity. It is important to note that: when $\delta t$ is small enough, the number of event cells $N$ tends to infinity. At this point, the entire cell sequence appropriately represents the raw event stream, while simultaneously fulfilling the input requisites for the neural network.

## 4.2 ADAPTIVE EVENT SLICING PROCESS

Recalling the event camera principle (Eq. 1) and spiking neuron mechanism (Eq. 3), both are brain-inspired processes that continuously respond to events/spikes upon reaching a threshold ($C/V_{th}$), and are fundamentally consistent as well as brain-inspired. This motivates us to employ spiking neural networks to efficiently process event streams.

Hence, we propose BEEF-Net, an SNN-based event processer, aiming for dynamically slicing the event stream that can satisfy our pre-set expectations, e.g., recognition accuracy improvement. Incorporating with BEEF-Net, we now describe the adaptive event slicing process:

Considering an event stream $\mathcal{E}$, we first convert $\mathcal{E}$ into a list of time-continuous event cells. We choose $\mathcal{F}$ as sum operation to aggregate cell into $C[n] \in \mathbb{R}^{2 \times H \times W}$. Event cells are then continuously entered into BEEF-Net through a loop operation. During forward propagation, the features of the last hidden layers ($h^{L-1}$) are finally mapped to a single spiking neuron to activate spikes:

$$S_{out} = \text{LIF}(\text{SNN}_{\text{FC}}(h^{L-1})). \tag{4}$$

Once the spiking neuron generates a spike (*i.e.*, $S_{out} = 1$) at time $n_s$, we immediately accumulate the event cells from the time following the last spike until the current spike to get a new dynamic cell: $\mathcal{D} = \sum_{n=0}^{n_s} \mathcal{C}[n]$.

**Event Slicing Rule:** The slice of the event stream is determined by the state (excited/resting) of the BEEF-Net's spiking neuron. Serving as a dynamic event trigger, BEEF-Net promptly decides to split events upon spike generation; conversely, it maintains continuous event input.

## 4.3 SPIKING POSITION-AWARE LOSS

In this section, we propose the Spiking Position-aware Loss (SPA-Loss), which contains two parts: (1) membrane potential-driven loss (Mem-Loss) is used to directly guide the spiking state of the spiking neuron at a specified timestamp , and (2) linear-assuming loss (La-Loss), which is designed to resolve the dependence phenomenon between neighboring membrane potentials, allowing the

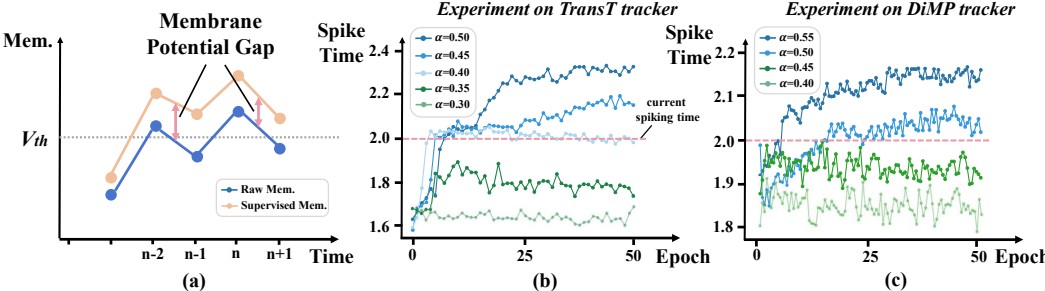

Figure 2: (a) 'Hill effect' in adaptive slicing process; (b) Impact of hyperparameter $\alpha$ settings on TransT tracker (Chen et al., 2021) and (c) DiMP tracker (Bhat et al., 2019).

neuron to fire spike at a more precise time. Moreover, we introduce a (3) dynamic hyperparameter tuning method to avoid the experimental bias caused by the manual setting of hyperparameters.

### 4.3.1 MEMBRANE POTENTIAL-DRIVEN LOSS

As mentioned in the previous section, the slice position of event is determined upon the spike occurrence. The challenge now lies in directing the SNN to trigger a spike precisely at the optimal position, once the label for this optimal slicing position is provided (in Sec. 4.4). Consider consecutive event cells as inputs starting from the previous spiking time, suppose we expect SNN to slice the event at $n^*$, *i.e.*, a spike $S_{out}$ is triggered at $n^*$. This corresponds to the membrane potential of the spiking neuron needing to reach the threshold $V_{th}$ at $n^*$, which inspired us to guide the spike time by directly giving the desired membrane potentials. However, membrane potential returns to the resting state immediately after the occurrence of a spike, which may result in inaccurate guidance at later moments (Appendix F). Thus, we choose to supervise the no-reset membrane potential $U[n]$ (Eq. 16). The membrane potential-driven loss is defined as:

$$\mathcal{L}_{Mem} = ||U[n^*] - (1 + \alpha)V_{th}||_2^2, \tag{5}$$

where $\alpha \geq 0$ is a hyper-parameter to control the desired membrane potential to exceed the threshold. However, an excessively high $\alpha$ may directly induce a premature spike in the neuron, thereby influencing the membrane potential state at the targeted time step. We provide a proposition to address this problem:

**Proposition 1.** *Suppose the input event cell sequence has length $N$, desired spiking time is $n^*$ ($n^* \in \{0, 1, ..., N\}$), the membrane potential at time $n^*$ satisfying the constraints:*

$$V_{th} \leq U[n^*] \leq \max(\beta V_{th} + \gamma I[n^*], V_{th}), \tag{6}$$

*where $I[n^*]$ is the input synaptic current from Eq.2. Then the spiking neuron fires a spike at time $n^*$ and does not excite spikes at neighboring moments.*

The proof is provided in the Appendix. Based on the proposition, we modify the loss function into:

$$\mathcal{L}_{Mem} = ||U[n^*] - ((1 - \alpha)U_{\text{lower}} + \alpha U_{\text{upper}})||_2^2, \tag{7}$$

where $U_{\text{lower}} = V_{th}$ and $U_{\text{upper}} = \max(\beta V_{th} + \gamma I[n^*], V_{th})$ denote the lower and upper bounds of the $U[n^*]$, respectively; $\alpha \in [0, 1]$ balances the desired membrane potential $U[n^*]$ between $U_{\text{lower}}$ and $U_{\text{upper}}$ (Appendix H). Experiments in Sec. 5.1 demonstrate that Mem-Loss is able to supervise the BEEF-Net to determine the slicing of the event flow at a specified timestamp.

### 4.3.2 LINEAR-ASSUMING LOSS

However, only using Mem-Loss is unable to guarantee that the spiking neuron can trigger spikes at any expected timestamp. We have the following observations.

**Observation 1** (Hill effect). Suppose there exists a situation where $S[n] = 1$ and $U[n] \geq U[n+1]$. If the neuron is expected to activate a spike at time $n+1$, $U[n+1]$ will be driven to reach the threshold through the Mem-Loss. Nonetheless, the supervised neuron still exhibits $U[n] \geq U[n+1]$, causing an early spike at time $n$.

As illustrated in Fig. 2(a), if $U[n] \geq U[n+1]$ exists, this membrane potential gap is still inherited after the supervision. In this case, when the later membrane potential is guided to exceed the threshold, the earlier membrane potential reaches the threshold sooner and turns the neuron into the resting state. This poses challenges in obtaining a second spike at the later moment.

Therefore, we expect the later membrane potential to increase monotonically with the time step to reverse the mountain range effect. Here we use the simplest linear monotonically increasing assumption to construct the loss function:

$$\mathcal{L}_{La} = \begin{cases} ||U[n_s] - V_{th} \frac{n_s}{n^*}||_2^2, & \text{if } U[n_s] \geq U[n^*] \text{ and } n_s < n^*; \\ 0, & \text{otherwise.} \end{cases} \tag{8}$$

We expect the membrane potential at $n_s$ to reach $\frac{n_s}{n^*} V_{th}$ in order for the latter membrane potential at the $n^*$ to reach $V_{th}$ in a linearly increasing form. More explanations are provided in Appendix H.1.

Combined with Mem-Loss and La-Loss, we defined the SPA-Loss, which guides the adaptive event slicing process in subsequent experiments:

$$\mathcal{L}_{SPA} = \mathcal{L}_{Mem} + \mathcal{L}_{La}. \tag{9}$$

### 4.3.3 DYNAMIC HYPERPARAMETER TUNING

Although controlling the SNN to spike at a desired location can be achieved through the combination of Mem-Loss and La-Loss, the utilization of varying $\alpha$ values (Eq. 7) may result in significant fluctuations in experimental results. We have the following observation.

**Observation 2.** The larger the $\alpha$, the earlier the BEEF-Net tends to fire spikes; and vice versa.

A larger $\alpha$ in Eq. 6 implies a higher pre-momentary membrane potential, which results in an earlier spike. Taking the larger-$\alpha$ scenario in Fig. 2(b), if BEEF-Net is expected to activate a spike at a later time, the larger $\alpha$ prevents the actual spike from being delayed. Consequently, we need to decrease $\alpha$, causing the expected spike time to shift earlier. This concludes that the update direction of $\alpha$ should be consistent with the update direction of the desired spiking index.

**Observation 3.** A fixed $\alpha$ leads to significant variations in performance across different tasks.

As illustrated in Fig. 2(b) and (c), the same $\alpha$ varies significantly on different downstream models, which makes it difficult to set the hyperparameter alpha in advance. Therefore, we should dynamically tune the value of $\alpha$ based on different tasks.

To address the above issues, we design a dynamic hyper-parameter tuning method and update $\alpha$ by Alg. 1. More details are provided in Appendix H.2.

### 4.4 FEEDBACK-UPDATE STRATEGY THROUGH SNN-ANN COOPERATION

Based on the methods proposed in the previous sections: if the desired trigger time $n^*$ is given, BEEF-Net is able to accurately accomplish the event slicing under the guidance of the SPA-Loss function. However, there still exists a problem that the conventional event-to-frame approach is independent of downstream performance. Hence, we propose a feedback-update strategy that enables the SNN to slice events when the downstream ANN model achieves optimal performance. By receiving real-time feedback from the downstream model and updating $n^*$, this strategy ultimately enhances task performance.

Particularly, when BEEF-Net triggers a spike at time $n_s$, it generates a spike output sequence $S = [0, ...0, 1, 0, ...]$, where 1 is at $n_s$-th. We first perform a *neighborhood search* to obtain $2d + 1$ candidate event cells within range $[n_s - d, n_s + d]$: $\{C[n_s - d], ..., C[n_s + d]\}$, where $C[n_s + i] = \sum_{n=0}^{n_s+i} C[n]$. We then choose a downstream model $\mathcal{M}$ (*e.g.*, object tracker or recognizer) and input candidate event cells into it to obtain feedback $y_k$:

$$y_k = \mathcal{M}(C[n_s - d]) \oplus ... \oplus \mathcal{M}(C[n_s + d]), \tag{10}$$

where $\mathcal{M}(C[n_s - i])$ returns the model output loss and $\oplus$ concatenates these losses into $y_k \in \mathbb{R}^{2d+1}$. We choose the model loss as the feedback since it directly reflects the quality of inputs. We can then

---

**Algorithm 1:** Feedback-Update Training Strategy

---

**Input:** SNN model, pretrained ANN model, ANN training dataset $\mathcal{D}_A$, SNN training dataset $\mathcal{D}_S$, training epoch $E_{train}$, epoch to
     start ANN finetuning $e_f$, $\alpha$ learning rate $\eta$.
**for** *all $e = 1, 2, \ldots E_{train}$ epoch* **do**
    **for** *all event batch $d_i = d_1, d_2, \ldots d_{N_S}$ in $\mathcal{D}_S$* **do**
        Feed $d_i$ along the time axis into SNN until it spikes at time step $n_s$;            // Eq. 4
        Generate event candidates and feed into ANN to get feedback;            // Eq. 11
        Calculate loss function $\mathcal{L}_{SPA} = \mathcal{L}_{Mem} + \mathcal{L}_{Mono}$;          // Eq. 9
        Backpropagate and update SNN parameters;
    **end**
    $\alpha \leftarrow \alpha - 2 \cdot \eta \sum_i^{N_S} (n^{*i} - n_s^i)/N_s$;            // Update $\alpha$ per epoch
    **if** $e > e_f$:
        Split $\mathcal{D}_A$ into $\mathcal{D}'_A$ by adaptive event slicing process based on SNN;        // Sec. 4.2
        Finetune ANN parameters on $\mathcal{D}'_A$;
**end**

---

generate the desired spike index $n^*$ through:

$$n^* = \arg \max_i f_{\text{norm}}(y_k[I]), \tag{11}$$

where the feedback is normalized by max-min normalization $f_{\text{norm}}$; $\arg \max$ extracts the index with the best feedback, which in turn guides the dynamic slicing process using SPA-Loss. At the same time, the ANN is being updated by feeding the new events after slicing, thus forming a continuous SNN-ANN cooperation process. We summarize the feedback-update strategy as follows:

**Feedback-Update Strategy.** The ANN model provides real-time feedback to the adaptive event slicing process, guiding BEEF-Net towards the optimal cutting point for the input events. Simultaneously, the ANN fine-tunes itself based on new events, creating a dynamic SNN-ANN update loop: SNN slicing $\xrightarrow{\text{new events}}$ ANN updating $\xrightarrow{\text{feedback}}$ SNN updating & slicing $\xrightarrow{\text{new events}}$ .... This strategy establishes a connection between raw data and the downstream model.

## 5 EXPERIMENTS

To evaluate the effectiveness of our proposed method, we set up two-level experiments. In the beginner's arena, we expect the SNN to find the exact slicing time with the simulated event inputs. In the expert's arena, we conduct experiments on event-based single object tracking (SOT) on the FE108 dataset (Zhang et al., 2021) and image recognition on DVS-Gesture (Amir et al., 2017) and N-Caltech101 (Orchard et al., 2015). Details of experiment settings are presented in the Appendix.

### 5.1 BEGINNER'S ARENA: LEARNING EVENT SLICING IN SIMPLE TASKS

We first conduct some entry-level tasks in order to validate the effectiveness of SPA-Loss. We set up the task: *Input $N$ randomized event cells, expect BEEF-Net to slice at a specified time step $n^*$ and there exists a certain probability of interfering with BEEF-Net to slice at other time steps.*

We compare our loss function with common mean square error (MSE-Loss) and cross-entropy loss (CE-Loss). We set the time step within the range $[1, 30]$ and the max number of iterations as 800. We utilize a lightweight convolutional SNN (2.02M) with random initialization. As depicted in Fig 3(a), our proposed SPA-Loss successfully supervises the SNN to activate spikes at the desired time steps. In particular, SPA-Loss requires only a small number of iterations ($<400$) to supervise the SNN to fire spikes at desired time steps. In contrast, MSE-Loss can only achieve supervision at a certain time step, and CE-Loss cannot even accomplish the task. In addition, using both Mem-Loss and La-Loss yields smoother results compared to using Mem-Loss alone. To summarize, the beginner's arena preliminarily tests the effectiveness of our proposed loss functions (including Mem-Loss and La-Loss) and paves the way for subsequent experiments.

### 5.2 EXPERT'S ARENA: MASTERING ADAPTIVE EVENT SLICING WITH SNN-ANN COLLABORATION

After a successful challenge in the beginner's arena, we move on to the expert arena. Here we use BEEF to adaptively process the event data on complex downstream tasks:

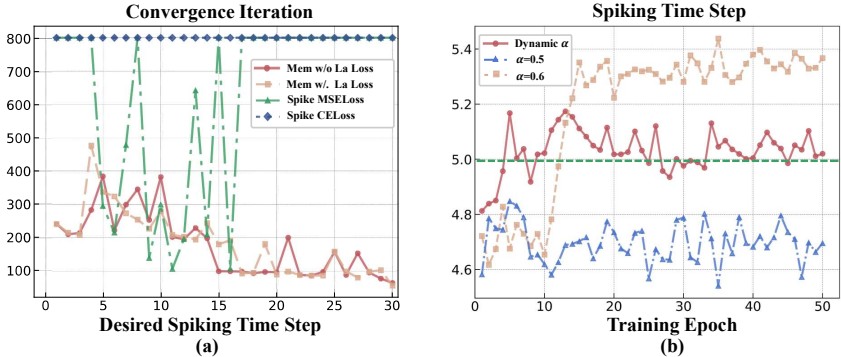

Figure 3: (a) Experiments on comparing different loss functions on a simple event slicing task. Our proposed Mem-Loss and La-Loss require only a small number of iterations to supervise the SNN to activate spikes at the desired time steps; (b) Experiments on different hyperparameter settings. Our dynamic tuning method can stably converge towards the optimal spiking time. In contrast, using a fixed $\alpha$ results in unstable training and challenges in finding the optimal point.

**Event-based Object Tracking.** Since the tracking task is highly sensitive to temporal information, dynamic event slicing is of great importance. Tab. 3 shows the quantitative comparisons of different state-of-the-art trackers. It can be observed that the tracking performances under the BEEF method have a significant improvement in terms of representative success rate (RSR), representative precision rate (RPR), and overlap precision (OP). For instance, TransT 's performance on $OP_{0.50}$ and $OP_{0.75}$ improved by 4.7% and 8.4% compared to its original result under the low light scenario. When compared to results on a fixed dataset (the number of fixed datasets is aligned with the number of datasets dynamically cut using BEEF to ensure a fair comparison), our method achieves favorable gains in the overall RSR, *i.e.*, 62.4 *vs.* 51.0.

**Event-based Recognition.** We also conduct experiments in event-based recognition to evaluate the effectiveness of our proposed method. As depicted in Tab. 1, our method has a significant improvement over the fixed-sliced method, with an accuracy improvement of 2.78% and 6.46% by using ResNet-34 in DVS-Gesture and N-Caltech101, respectively. To verify that the results of our adaptive slicing method are not

Table 1: Quantative comparison on DVS-Gesture and N-Caltech101. *random* and *fix* denote that the input event frames are randomly sliced and fixed sliced, respectively. Instead, our BEEF adaptively sliced the event stream into frames.

| Method | DVS-Gesture | | | N-Caltech101 | | |
|---|---|---|---|---|---|---|
| Input Type | Random | Fix | **Ours** | Random | Fix | **Ours** |
| ResNet-18 | 93.06 | 93.40 | **94.79** | 77.80 | 73.37 | **79.86** |
| ResNet-34 | 95.14 | 93.40 | **96.18** | 78.77 | 76.08 | **82.54** |

biased due to randomness, we add the random-slice baselines for comparison, in which the event stream is randomly sliced into event frames and fed into the ANN for training. Our method also yields better performance compared with the random-slice results.

**Visualization of Adaptive Event Slicing.** We visualize the tracking results to demonstrate that the dynamic slicing method is able to adapt to various motion scenarios. As shown in Fig. 9, our method obtains better tracking performance compared to fixed event inputs, *i.e.*, the position of the prediction box is more accurate. Additionally, our dynamic event slicing method can achieve (1) *edge enhancement* and (2) *redundancy removal* to refine the event input under different tracking scenarios. However, the fixed-slice approach adopts the same slicing strategy for each event stream, leading to performance degradation.

**Analysis of Energy Consumption and Processing Speed.** We evaluated the energy consumption and processing speed on GPU for both the SNN and ANN Tracker models during dynamic event slicing (detailed method in Appendix J). As illustrated in Tab. 2, compared to the 35.926 $mJ$ required by the ANN Tracker. Moreover, the SNN model has a significantly

Table 2: Comparison of Efficiency and Speed. ACs represent accumulated spike operations, while MACs indicate multiply-accumulate operations. FPS metrics are based on GPU processing.

| Models | ACs (G) | MACs (G) | Energy (mJ) | FPS (Hz) |
|---|---|---|---|---|
| SNN | 0.0059 | 0.0036 | 0.022 | 111 |
| ANN Tracker | 0 | 7.81 | 35.926 | 39 |

lower energy consumption at 0.022 $mJ$ per image, and processes at a speed of 111 FPS on the GPU, which is approximately 2.85 $\times$ faster than the ANN Tracker's speed. Consequently, our BEEF model can facilitate real-time dynamic event slicing with negligible energy consumption.

Table 3: Quantitative comparison on FE108 in terms of representative success rate (RSR), representative precision rate (RPR), and overlap precision (OP). There are four challenging scenarios, including high dynamic range (HDR), low light (LL), fast motion with and without motion blur (FWB & FNB) and all testing datasets (ALL). [*method*]+*BEEF* represents the results based on our adaptive event to frame framework and the results of [*method*] are tested on the original fixed-sliced event dataset from (Zhang et al., 2021). To ensure a fair comparison, *fix event* indicates that the model is tested on a dataset of fixed-sliced event frames, where the number of fixed event frames is the same as the number of dynamically sliced event frames by using BEEF.

| Methods | HDR | | | | LL | | | | FWB | | | | FNB | | | | ALL | | | |
|---|---|---|---|---|---|---|---|---|---|---|---|---|---|---|---|---|---|---|---|---|
| | RSR | OP$_{.50}$ | OP$_{.75}$ | RPR | RSR | OP$_{.50}$ | OP$_{.75}$ | RPR | RSR | OP$_{.50}$ | OP$_{.75}$ | RPR | RSR | OP$_{.50}$ | OP$_{.75}$ | RPR | RSR | OP$_{.50}$ | OP$_{.75}$ | RPR |
| SiamRPN (Li et al., 2018) | 15.3 | 16.9 | 6.1 | 21.6 | 10.1 | 8.3 | 1.4 | 14.5 | 26.2 | 32.1 | 6.1 | 44.1 | 33.2 | 42.9 | 11.5 | 51.9 | 21.8 | 26.1 | 7.0 | 33.5 |
| ATOM (Danelljan et al., 2019) | 36.6 | 41.8 | 14.4 | 56.0 | 28.6 | 29.1 | 5.8 | 45.0 | 66.8 | 89.6 | 32.6 | 96.7 | 57.1 | 71.0 | 28.0 | 88.6 | 46.5 | 56.4 | 20.1 | 71.3 |
| SiamFC++ (Xu et al., 2020) | 15.3 | 15.0 | 1.3 | 25.2 | 13.4 | 8.7 | 0.8 | 15.3 | 28.6 | 36.3 | 6.0 | 48.2 | 36.8 | 42.7 | 7.4 | 63.1 | 23.8 | 26.0 | 3.9 | 39.1 |
| SiamBAN (Chen et al., 2020) | 16.3 | 16.4 | 3.9 | 26.6 | 15.5 | 14.8 | 2.3 | 26.5 | 25.2 | 26.3 | 5.8 | 46.7 | 32.0 | 39.6 | 9.1 | 51.4 | 22.5 | 25.0 | 5.6 | 37.4 |
| KYS (Bhat et al., 2020) | 15.7 | 14.5 | 5.2 | 23.0 | 12.0 | 8.0 | 1.1 | 18.0 | 47.0 | 63.9 | 14.8 | 73.3 | 36.9 | 44.5 | 15.2 | 57.9 | 26.6 | 30.6 | 9.2 | 41.0 |
| CLNet (Dong et al., 2020) | 30.0 | 33.5 | 9.6 | 48.3 | 13.7 | 6.0 | 0.9 | 23.6 | 52.9 | 71.2 | 23.3 | 80.3 | 40.8 | 46.3 | 14.2 | 67.7 | 34.4 | 39.1 | 11.8 | 55.5 |
| DiMP (Bhat et al., 2019) | 41.8 | 50.0 | 17.9 | 62.7 | 45.6 | 52.8 | 11.2 | 69.5 | 69.4 | 94.7 | 37.1 | 99.7 | 60.5 | 75.6 | 29.3 | 93.2 | 52.6 | 65.4 | 23.4 | 79.1 |
| DiMP (fixed event) | 53.3 | 68.2 | 21.4 | 81.6 | 67.6 | 86.3 | 43.1 | 95.0 | 49.7 | 45.4 | 5.88 | 80.5 | 49.6 | 59.4 | 23.7 | 75.3 | 53.8 | 64.3 | 19.5 | 82.4 |
| DiMP+**BEEF** | 52.3 | 65.5 | 21.4 | 78.7 | 69.4 | 92.4 | **43.2** | 97.5 | 52.4 | 64.8 | 26.6 | 78.8 | **62.2** | **81.0** | 21.8 | **97.2** | 56.0 | 71.4 | 23.3 | 85.1 |
| PrDiMP (Danelljan et al., 2020) | 44.3 | 52.8 | 19.6 | 66.3 | 44.6 | 48.2 | 8.9 | 69.5 | 67.0 | 89.9 | 33.6 | 96.7 | 60.6 | 75.8 | 29.7 | 93.3 | 53.0 | 65.0 | 23.3 | 80.5 |
| PrDiMP (fixed event) | 41.2 | 49.8 | 18.4 | 66.1 | 42.7 | 45.2 | 12.5 | 87.1 | 62.4 | 85.8 | 21.3 | 90.4 | 47.6 | 58.3 | 18.5 | 77.0 | 48.0 | 61.2 | 19.2 | 78.6 |
| PrDiMP+**BEEF** | 49.1 | 59.0 | 15.6 | 76.0 | 69.7 | **94.0** | 41.8 | 94.7 | 63.2 | 86.0 | 22.4 | 91.7 | 51.7 | 60.4 | 19.7 | 81.6 | 54.9 | 68.4 | 19.8 | 83.9 |
| TransT (Chen et al., 2021) | 55.9 | 71.0 | 24.6 | **84.5** | 66.8 | 88.9 | 34.3 | 96.5 | 74.1 | **98.6** | 54.0 | **99.9** | 55.8 | 69.2 | 24.9 | 85.4 | 59.6 | 76.4 | 29.0 | 88.8 |
| TransT (fixed event) | 51.4 | 67.8 | 11.1 | 81.2 | 63.2 | 80.2 | 28.3 | 89.3 | 41.5 | 28.0 | 2.50 | 57.7 | 50.6 | 57.9 | 12.7 | 78.9 | 51.0 | 59.0 | 12.0 | 78.8 |
| TransT+**BEEF** | **57.7** | **75.2** | **28.1** | 82.6 | **70.7** | 93.6 | 42.7 | **99.0** | **74.9** | 97.7 | **61.1** | 98.6 | 58.7 | 75.6 | **29.6** | 84.6 | **62.4** | **81.2** | **34.1** | **88.9** |

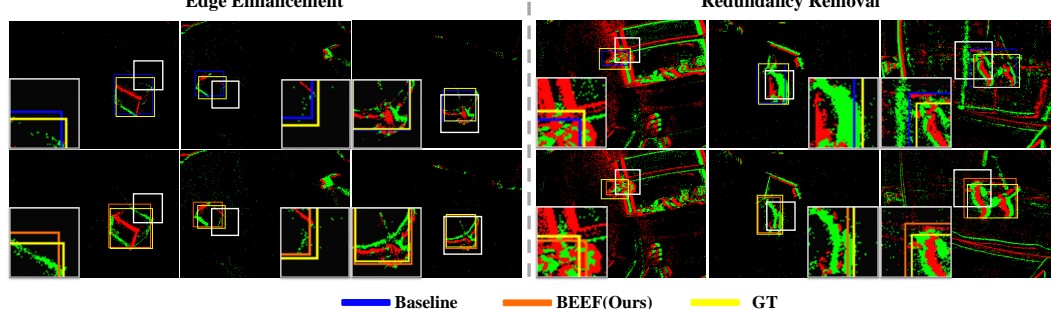

**Edge Enhancement**   **Redundancy Removal**

— Baseline   — BEEF(Ours)   — GT

Figure 4: Visualization on FE108 dataset. The white box denotes the zoom-in area. Our adaptive event slicing method provides better tracking performance than fixed counterparts while enabling edge enhancement and redundancy removal.

### 5.3 EVALUATION OF THE DYNAMIC HYPERPARAMETER TUNING

As described in Sec. 4.3.3, the dynamic hyperparameter tuning method can solve the problems of spiking moment offset and maintain experimental stability. Here we use different $\alpha$ settings in tracking experiments to study the effect of hyperparameters. Fig. 3 shows that the fixed $\alpha$ setting does not allow the spiking time step to reach the desired time step accurately, while our dynamic $\alpha$ tuning allows BEEF-Net to spike at the desired time step, leading to a better event-slicing process.

## 6 CONCLUSION AND FUTURE WORK

In this work, we proposed BEEF, a novel event processing framework that adaptively splits event streams into frames. BEEF utilized a spiking neural network (SNN) as an event trigger, which determines the appropriate slicing time according to the generated spikes. To achieve accurate slicing, we designed the Spiking Position-aware Loss (SPA-Loss) function which guided the SNN to trigger spikes at the desired time step by controlling the membrane potential value. In addition, we proposed a Feedback-Update training strategy that allows the SNN to make accurate event slicing decisions based on the ANN feedback. Extensive experiments have shown the effectiveness of BEEF in achieving top-tier performance in event-based object tracking and recognition tasks. In the future, we will assess BEEF's suitability for other event-based tasks, and devise more efficient training strategies for the SNN-ANN cooperative framework to optimize real-time processing.

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

# Appendix

## A DETAILS OF OUR MOTIVATIONS

To clarify the motivation behind our dynamic event stream slicing algorithm, this section tells the details.

### A.1 MOTIVATION FOR PROPOSING A DYNAMIC EVENT STREAM SLICING ALGORITHM

The process of event-to-frame conversion is mainly divided into two steps: **Step 1. Slice the raw event stream into multiple sub-event stream**, and **Step 2. convert these sub-event streams into frames using various event representation methods.** While much work has focused on optimizing event representation (Step 2) to extract better event information, including time surface and EST, they do not address the issues arised with fixed slicing (*e.g.*, resulting non-uniform event in scenarios with changing motion speed). Despite event slicing being a small part of the overall pipeline, it is a critical point. This is because the event stream is very sensitive to slicing, and the model performance fluctuates very much for different slicing methods, as proved by extensive experiments in Appendix B.

To better address this issue, we introduced the dynamic slicing framework BEEF. Meanwhile, BEEF is guided by downstream task feedback to ensure that the new sub-streams could enhance downstream task performance.

### A.2 MOTIVATION FOR FOR USING SNN AS A SLICING TRIGGER

The reason why we choose SNN as the event slicing trigger is twofold:

- Utilizing SNNs on neuromorphic hardware for processing event streams is low-energy and low-latency (Davies et al., 2018; Akopyan et al., 2015).
- Deployed on neuromorphic hardware, SNNs can process event streams asynchronously (Roy et al., 2023; Viale et al., 2021; Yu et al., 2023), conserving energy when there is no data input—a capability that GPUs, operating synchronously, lack.

Due to the aforementioned reasons, there is a considerable amount of research (Hagenaars et al., 2021; Yao et al., 2021; Zhu et al., 2022) employing Spiking Neural Networks (SNNs) for event data. Although these SNNs are simulated on GPU platforms, the models resulting from such simulations could be deployed on neuromorphic hardware.

### A.3 CONTRIBUTION FOR USING SNN AS A SLICING TRIGGER

We propose a new cooperative paradigm where SNN acts as an efficient, low-energy data processor to assist the ANN in improving downstream performance. This is a brand-new SNN-ANN cooperation way, paving the way for future event-related implementation on neuromorphic chips.

## B SENSITIVITY ANALYSIS OF FIXED EVENT SLICING METHOD

To demonstrate that events are sensitive to slicing by fixed methods, and to emphasize the importance of proposing a dynamic event slicing approach, we have conducted a total of 60 experiments with different models to investigate the impact of different slicing techniques and different numbers of slices on the performance in downstream tasks.

In our experiment, we employed two fixed slicing methods: (1). *Slicing with a fixed number of events*, and (2). *Slicing with a fixed duration*. $N$ denotes the number of resulting event slices. Experimental results are detailed as follows in Tab. 4 and Fig. 5.

The results indicate significant fluctuations (large variance) in downstream performance based on the slicing method and the number of slices used. We believe this addition effectively demonstrates the sensitivity of event streams to slicing techniques, **confirming the need for our motivation to**

| N-Caltech101 | $N$ | 2 | 4 | 6 | 8 | 10 | 12 | 14 | 16 | 18 | 20 | 22 | 24 | 26 | 28 | 30 | Mean | Var |
|---|---|---|---|---|---|---|---|---|---|---|---|---|---|---|---|---|---|---|
| ResNet18 | Fixed Count | 70.96 | 75.26 | 75.39 | 75.30 | 76.09 | 73.95 | 74.09 | 73.80 | 76.40 | 75.39 | 75.45 | 73.60 | 71.94 | 71.01 | 71.17 | **73.98** | **3.33** |
| ResNet18 | Fixed Time | 62.90 | 72.64 | 76.38 | 74.48 | 74.91 | 73.70 | 74.30 | 74.69 | 76.95 | 74.75 | 74.46 | 74.42 | 71.61 | 71.52 | 69.69 | **73.16** | **10.80** |
| ResNet34 | Fixed Count | 72.19 | 75.55 | 76.98 | 78.22 | 77.14 | 77.40 | 76.78 | 76.90 | 78.14 | 77.06 | 76.91 | 74.85 | 74.76 | 76.91 | 73.07 | **76.19** | **2.90** |
| ResNet34 | Fixed Time | 65.42 | 75.92 | 78.29 | 78.20 | 78.48 | 76.22 | 77.76 | 76.57 | 75.94 | 76.80 | 76.61 | 75.91 | 75.11 | 74.76 | 74.19 | **75.74** | **9.15** |

Table 4: The sensitivity analysis of fixed event slicing on N-Caltech101. The results demonstrate that the event is sensitive to the fixed slicing method (slicing by fixed time or event count), thereby affirming the need for proposing a dynamic slicing method.

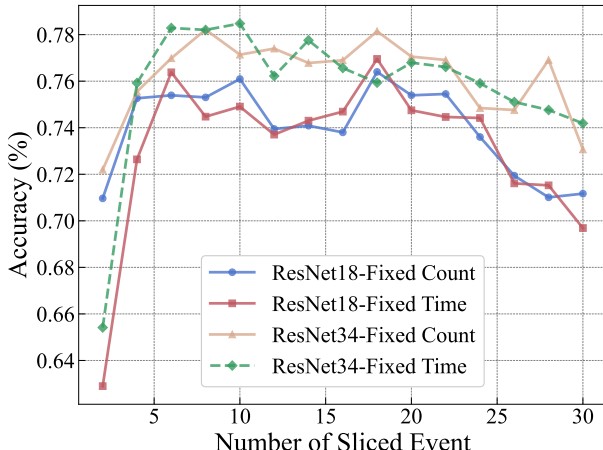

Figure 5: Visualization of sensitivity analysis on N-Caltech101 dataset. The fluctuations in accuracy for different numbers of sliced event with different fixed slicing methods are significant, demonstrating that events are very sensitive to fixed slicing methods.

**propose dynamic slicing of event streams.** Additionally, the accuracy achieved using the dynamic slicing method (82.54% by ResNet34) surpasses that of any fixed slicing approach (with the highest being 78.48%), further substantiating the efficacy of the dynamic method in our study.

## C  ILLUSTRATION OF BEEF VS. FIXED SLICING METHOD

In order to more intuitively show the difference between our dynamic event slicing method and the traditional fixed slicing method, we specifically illustrates these methods through Fig. 6.

Fig. 6 (a) denotes that each resulting sliced event has the same duration, and Fig. 6 (b) denotes that the number of event points contained in each resulting sliced event is the same. Since event stream are usually unevenly distributed, a fixed cutting method often leads to non-uniform event information (*e.g.*, in scenarios with changing motion speed). In contrast, our approach decides the optimal slice position through feedback from downstream ANN by using an SNN as the event slicing trigger.

## D  DIFFERENCE BETWEEN EVENT SLICING AND EVENT REPRESENTATION

It is worth noting that **our work focuses on the slicing of the event stream rather than focusing on event representation**. Event representation refers to the process of event information extraction that is performed after the event stream has been sliced into sub-event stream, and the resulting event representation meets the neural network input requirements. Thus, our dynamic slicing process and event representation can be used at the same time, either better slicing or representation method benefits the feature extraction with neural network, thus improving performance.

To validate the effectiveness of our slicing approach, we supplement the event-based recognition task below. We compare the downstream performance of three different event representation methods

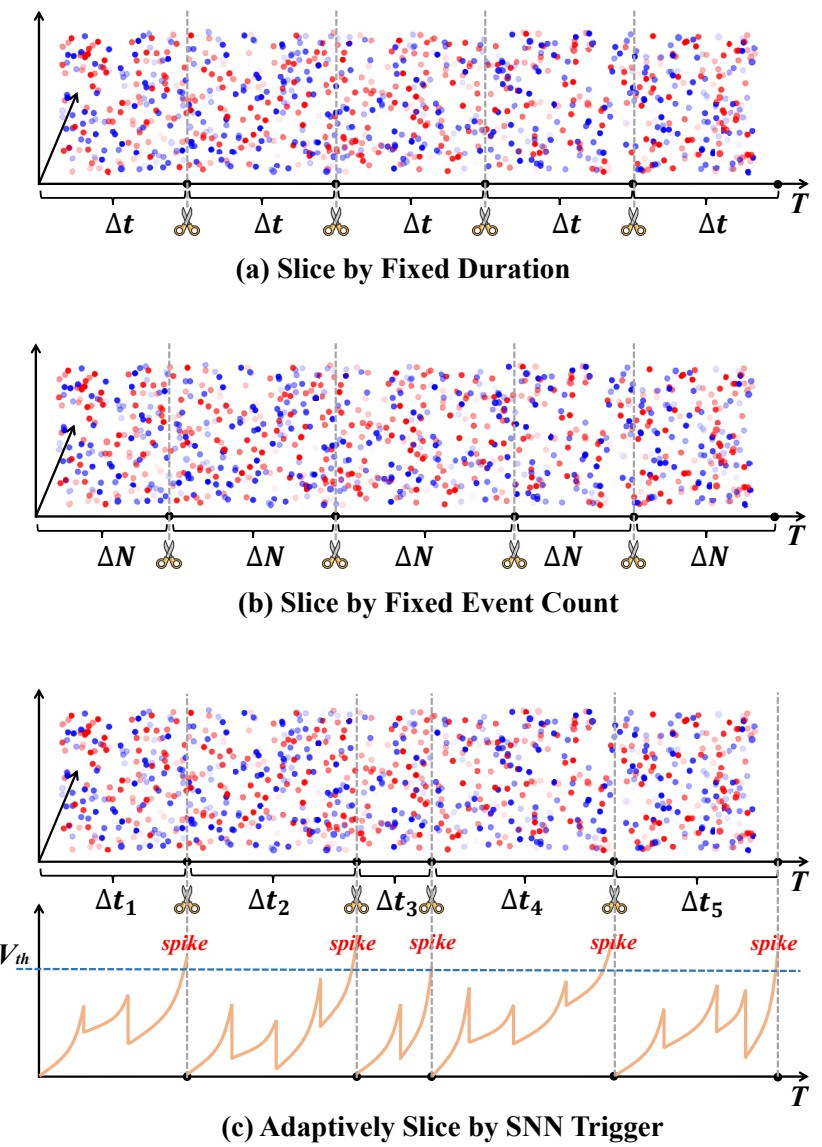

(a) Slice by Fixed Duration

(b) Slice by Fixed Event Count

(c) Adaptively Slice by SNN Trigger

Figure 6: Illustration of our adaptive slicing method BEEf compared with traditional fixed slicing methods. Both (a) slicing by fixed duration and (b) slicing by fixed event count are traditional event slicing methods. Our BEEF (c) can adaptively decides the optimal slicing position through ANN-SNN cooperation.

(including Event Frame (Maqueda et al., 2018), Event Spike Tensor (EST (Gehrig et al., 2019)) and Voxel Grid (Zhu et al., 2019)) on the DVSGesture dataset under fixed slicing and dynamic slicing method in Tab. 5.

| DVSGesture | Event Frame | Event Spike Tensor | Voxel Grid |
|---|---|---|---|
| Fix Duration | 93.75% | 93.75% | 88.54% |
| Fix Event Count | 93.06% | 94.79% | 88.19% |
| **BEEF(ours)** | **94.79%** | **95.49%** | **89.24%** |

Table 5: Experiments on different event representations with fixed/dynamic slicing methods. Our BEEF yields significant improvement when using different event representation methods.

# E  DEFINITION OF GENERAL EVENT REPRESENTATION

Based on the definition of event field from (Gehrig et al., 2019), we here define a general version of event representation as a mapping $\mathcal{M} : \mathcal{E} \mapsto \mathcal{T}$ between the set $\mathcal{E}$ and a tensor $\mathcal{T}$:

**Definition 1** (General event representation). *Based on a measurement $\mathbf{1}_{\text{condition}}$, general event representations are grid-like tensors defined in continuous space and time:*

$$G_{\pm}(x, y, t, \text{condition}) = \sum_{e_k \in \mathcal{E}_{\pm}} \mathbf{1}_{\text{condition}}(x, y, t)\delta(x - x_k, y - y_k)\delta(t - t_k), \qquad (12)$$

where $\pm$ denotes the event polarity; $\mathbf{1}_{\text{condition}}$ sets the specific approach of representation to each event, *e.g.*, condition $= \{\sum_{e_k} = M\}$ denotes the number of event points in each grid-like tensor is fixed to $M$, *i.e.*, slicing event stream $\mathcal{E}$ by *event count*. $G_{\pm}$ are grid tensors with $x \in \{0, 1, ..., W - 1\}, y \in \{0, 1, ..., H - 1\}$, and $t \in \{t_0, t_0 + \Delta t, ..., t_0 + B\Delta t\}$, where $t_0$ is the first time stamp, $\Delta t$ is the bin size determined by the representation condition, and $B$ is the number of temporal bins. Eq. (12) converts raw event into grid-like representations by a Dirac pulse in the space-time manifold.

However, such event representation is imprecise due to the fact that the process of $\mathbf{1}_{\text{condition}}$ is fixed, leading to both spatial and temporal information loss. The main objective of this study is to solve the problem of fixed slicing of the event stream and to provide a dynamic segmentation scheme. To closely simulate the asynchronous input process of the event stream, we hence introduce the definition of event cell in Sec. 4.1.

# F  REASON FOR USING NO-RESET MEMBRANE POTENTIAL

We first recall the definition of original membrane potential $V[n]$:

$$V[n] = \beta V[n - 1] + \gamma I[n], \qquad (13)$$
$$S[n] = \Theta(V[n] - \vartheta_{\text{th}}), \qquad (14)$$
$$V[n] = V[n](1 - S[n]) + V_{\text{reset}}, \qquad (15)$$

where the the neuron will reset its membrane potential to $V_{\text{reset}} < \vartheta_{\text{th}}$ at time $n$ once it trigger a spike $S[n]$. As described in Sec. 4.3, we choose to guide the membrane potential without reset stage (Eq. 15) $U[n]$ (or named no-reset membrane potential), instead of the normal membrane potential $V[n]$. The no-reset membrane potential is defined similarly as $V[n]$:

$$U[n] = \beta U[n - 1] + \gamma I[n], \qquad (16)$$
$$S[n] = \Theta(U[n] - \vartheta_{\text{th}}), \qquad (17)$$

but the neuron does not reset its membrane potential in this condition. The reason behind this choice is that the reset process will affect the guidance of the $V[n]$. Specifically, suppose we expect the neuron to fire a spike at $n + 1$, but if a spike just occurs at time $n$, the membrane potential $V[n]$ will reset to $V_{\text{reset}}$, consequently leading to a small value for $V[n + 1]$. Based on the SPA-Loss function, $V[n + 1]$ would then be guided by a large expected membrane potential value (above the threshold). However, this would incorrectly guide the membrane potential after resetting to the desired membrane potential, rather than guiding the true membrane potential as intended. Therefore, we choose to use the no-reset membrane potential $U[n]$ to effectively guide the spiking neuron to fire spike at the specified location.

# G  PROOF OF PROPOSITION

**Proposition 1.** *Suppose the input event cell sequence has length $N$, desired spiking time is $n^*$ ($n^* \in \{0, 1, ..., N\}$), the membrane potential at time $n^*$ satisfying the constraints:*

$$V_{th} \leq U[n^*] \leq \max(\beta V_{th} + \gamma I[n^*], V_{th}), \qquad (18)$$

*where $I[n^*]$ is the input synaptic current from Eq.2. Then the spiking neuron fires a spike at time $n^*$ and does not excite spikes at neighboring moments.*

*Proof.* Here we consider two conditions that affect the spiking state at moment $n^*$:

(1) Membrane potential at time $n^*$ is too small to emit a spike.

(2) Membrane potential at time $n^*$ is too large, affecting neighboring moment spiking states.

To satisfy the condition (1), we only need to guide the membrane potential $U[n^*]$ to reach the threshold $V_{th}$ at time $n^*$, thus the upper bound of $U[n^*] = V_{th}$. In condition (2), we need to consider the state of the membrane potential at $n^* - 1$ and $n^* + 1$. We first exhibit the accumulation rules of membrane potential:

$$U[n^*] = \beta U[n^* - 1] + \gamma I[n^*] \tag{19}$$

However, if $U[n^*]$ is too large, this may cause the membrane potential $U[n^* - 1]$ to exceed the threshold and occur spike generation prematurely, and then the membrane potential will immediately drop to a reset value (Eq. 15). This will leave the membrane potential $U[n^*]$ at a very low value, making it difficult to trigger a spike. Hence, we should control the membrane potential not to exceed the threshold value at moment $n^* - 1$:

$$U[n^* - 1] = \frac{(U[n^*] - \gamma I[n^*])}{\beta} \leq V_{th} \tag{20}$$

$$\Rightarrow \quad U[n^*] \leq \beta V_{th} + \gamma I[n^*] \tag{21}$$

Thus, the upper bound of $U[n^*] = \beta V_{th} + \gamma I[n^*]$. However, if the leaky factor $\beta$ is small, there exists a possibility that $\beta V_{th} + \gamma I[n^*] \leq V_{th}$, thus we set the upper bound of $U[n^*]$ as $\max(\beta V_{th} + \gamma I[n^*], V_{th})$.

Next, we consider whether the spike at time $n^*$ affects the pulse state at time $n^* + 1$ Since the neuron at the $n^*$ moment has already completed the spike generation before accumulating $U[n^*+1]$. Therefore the membrane potential at $n^*$ does not affect the neuronal state at $n^* + 1$. In sum, if the membrane potential satisfies: $V_{th} \leq U[n^*] \leq \max(\beta V_{th} + \gamma I[n^*], V_{th})$, the spiking neuron fires a spike at moment $n^*$ and does not excite spikes at neighboring moments.

## H   MORE EXPLANATIONS IN MEMBRANE POTENTIAL-DRIVEN LOSS

In Sec. 4.3.1, we explore the range of the expected membrane potential $U[n^*]$ and ensure its rationality by proposition 1 (proved in Appendix. G). To understand the setting of the membrane potential more easily, we visualize the boundary cases below:

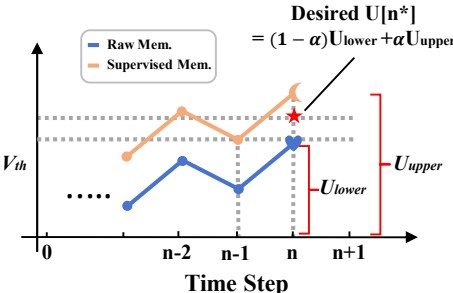

Figure 7: Visualization of the boundary cases when controlling the desired membrane potential, where the 'heart-like' point denotes the lower bound case and the 'moon-like' point denotes the upper bound case.

The lower bound case means that the membrane potential at the desired index should be at least $V_{th}$ to activate a spike, and the upper bound case guides the membrane potential to exceed the threshold but prevents generating spikes in the previous time step. Hence, the desired membrane potential should be bounded in $[V_{th}, \max(\beta V_{th} + \gamma I[n^*], V_{th})]$ and $\alpha \in [0, 1]$ (Eq. 7) balances the desired membrane potential $U[n^*]$ between $U_{\text{lower}}$ and $U_{\text{upper}}$.

### H.1   DETAILS IN LINEAR-ASSUMING LOSS

As described in Sec. 4.3.2, suppose we expect the SNN to trigger a spike at a later time, if there exists a hill effect, the earlier membrane potential always reaches the excited state sooner and turns

the neuron into the resting state to suppress the spiking generation at later moments. To address this challenge, we expect the later membrane potential to satisfy: (1) the later membrane potential should be larger than the current membrane potential to reverse the hill effect, and (2) the later membrane potential should exceed the threshold to fire a spike. Hence, we assume that the membrane potential in this condition should increase monotonically with the time step, as illustrated in Fig. 8.

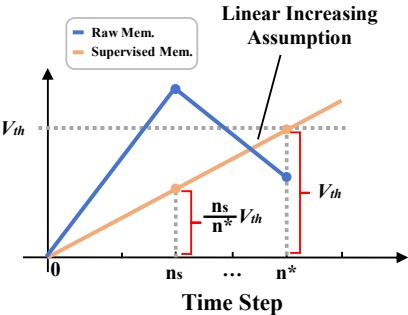

Figure 8: Visualization of our expected linearly increasing membrane potential.

Then we use the La-Loss to supervise the membrane potential at $n_s$ to reach $\frac{n_s}{n^*}V_{th}$ in order for the latter membrane potential at the $n^*$ to reach $V_{th}$, satisfying both (1) and (2).

### H.2 DETAILS OF DYNAMIC HYPERPARAMETER TUNING

To deeply explore the control of the hyperparameter $\alpha$, we first analyze the effect of $\alpha$ in Eq. 7. When $\alpha$ reaches its maximum value (*i.e.* $\alpha = 1$), the desired membrane potential evolves to $U_{upper}$, which corresponds to a situation where the membrane potential just reaches the threshold at the previous moment (in Fig. 7). That is, as $\alpha$ increases, the previous moment is more likely to generate a spike, driving the spiking time earlier. Recalling observation 2, taking a large-$\alpha$ scenario as an example, the SNN fails to spike at a later time step due to the alpha being too large, limiting the neuron's ability to spike later. Hence, we expect the $\alpha$ to decrease to allow the desired index to decrease as well; similarly for small $\alpha$ scenarios. To summarize, we hope the $\alpha$ is updated in the same direction as the desired index, we update $\alpha$ by setting:

$$\alpha.\text{grad} = ||\sum_{i}^{N_s}(n^{*i} - n_s^i)/N_s||_2^{2'}, \tag{22}$$

$$\alpha \leftarrow \alpha - 2 \cdot \eta \sum_{i}^{N_s}(n^{*i} - n_s^i)/N_s, \tag{23}$$

where $\text{grad}$ denotes the gradient of $\alpha$ and $'$ denotes derivative operation. The explanations of other math symbols can be found in the main text and Alg. 1.

## I IMPLEMENTATION DETAILS

We adopt a spiking neural network with structure: {*16C3-IF-AP2-32C3-IF-AP2-64C3-IF-AP2-LN-IF-LN-IF*}, which consists of three convolutional layers and 2 linear layers, no residual block or attention are used. We choose the IF neuron as the activation function. We adopt the SGD optimizer and set the initial learning rate as 1e-4, along with the cosine learning rate scheduler. SNN models are trained for 50 epochs with batch size 32.

### I.1 EVENT-BASED OBJECT TRACKING

**Datasets.** The FE108 dataset is an extensive event-based dataset for single object tracking, including 21 different object classes and several challenging scenes, *e.g.*, low-light (LL) and high dynamic range (HDR). The event streams are captured by a DAVIS346 event-based camera, which equips

a 346x260 pixels dynamic vision sensor (DVS). We choose 54 sequences for training ANNs, 22 sequences for training SNNs and the rest 32 sequences for testing.

**Evaluation Metrics.** To show the quantitative performance of each tracker, we utilize three widely used metrics: success rate (Suc.), precision rate (Prec), normed precision rate (N-Prec), and overlap precision (OP). These metrics represent the percentage of three particular types of frames. Success rate is the frame of that overlap between the ground truth and the predicted bounding box is larger than a threshold; Precision rate focuses on the frame of the center distance between ground truth and predicted bounding box within a given threshold; $OP_{thres}$ represents SR with $thres$ as the threshold. We employ the area under curve (AUC) to represent the success rate. The precision score is associated with a 20-pixel threshold.

## I.2 EVENT-BASED RECOGNITION

**DVS-Gesture.** The DVS-Gesture (Amir et al., 2017) dataset contains 11 hand gestures from 29 subjects under 3 illumination conditions, recorded by a DVS128.

**N-Caltech101.** The N-Caltech101 dataset (Orchard et al., 2015) incorporates 8,831 event-based images, with a 180×240 resolution and 101 classes, generated from the original Caltech101 dataset through an event-based sensor.

## J  THEORETICAL ENERGY CONSUMPTION CALCULATION

To calculate the theoretical energy consumption, we begin by determining the synaptic operations (SOPs). The SOPs for each block in the BEEF can be calculated using the following equation:

$$\text{SOPs}(l) = fr \times T \times \text{FLOPs}(l) \tag{24}$$

where $l$ denotes the block number in the BEEF, $fr$ is the firing rate of the input spike train of the block and $T$ is the time step of the spike neuron. $\text{FLOPs}(l)$ refers to floating point operations of $l$ block, which is the number of multiply-and-accumulate (MAC) operations. And SOPs are the number of spike-based accumulate (AC) operations.

To estimate the theoretical energy consumption of BEEF, we assume that the MAC and AC operations are implemented on a $45nm$ hardware, with energy costs of $E_{MAC} = 4.6pJ$ and $E_{AC} = 0.9pJ$, respectively. According to (Panda et al., 2020; Yao et al., 2023), the calculation for the theoretical energy consumption of BEEF is given by:

$$
\begin{aligned}
E_{\text{Diffusion}} = {} & E_{MAC} \times \text{FLOP}^1_{\text{SNN}_{\text{Conv}}} \\
& + E_{AC} \times \left( \sum_{n=2}^{N} \text{SOP}^n_{\text{SNN}_{\text{Conv}}} + \sum_{m=1}^{M} \text{SOP}^m_{\text{SNN}_{\text{FC}}} \right)
\end{aligned} \tag{25}
$$

where $N$ and $M$ represent the total number of layers of Conv and FC, $E_{MAC}$ and $E_{AC}$ represent the energy cost of MAC and AC operation, $\text{FLOP}_{\text{SNN}_{\text{Conv}}}$ denotes the FLOPs of the first Conv layer, $\text{SOP}_{\text{SNN}_{\text{Conv}}}$ and $\text{SOP}_{\text{SNN}_{\text{FC}}}$ are the SOPs of $n^{th}$ Conv and $m^{th}$ FC layer, respectively.

## K  MORE EXPERIMENTS ON BEGINNER'S ARENA

**Problem Setup.** To verify the accuracy of our proposed slicing method, we expect BEEF-Net can slice events at the specified time step. We set up two scenarios and only show the difficult task (II) in the main text:

- *Task (I): Input $T$ identical event cells, expect BEEF-Net to slice at a specified time step $T^*$.*
- *Task (II): Input $T$ randomized event cells, expect BEEF-Net to slice at a specified time step $T^*$, but there exists a certain probability of interfering with BEEF-Net to slice at other locations.*

Task (I) aims to verify whether our proposed slicing strategy can accurately locate the optimal point; To test the robustness of our method, task (II) simulates complex event stream processing with random inputs and adds random noise to affect BEEF-Net with wrong labels.

Table 6: Results on simple event slicing tasks with SPA-Loss.

|  | Input Size | Time Steps | Parameter | Iterations to Convergence ↓ |
|---|---|---|---|---|
| Task (I) | $32\times 32$ | 30 | 0.52M | 75 |
|  | $64\times 64$ | 30 | 2.02M | 81 |
| Task (II) | $32\times 32$ | 100 | 0.52M | 29 |
|  | $64\times 64$ | 100 | 2.02M | 88 |

We adopt a lightweight BEEF-Net (0.25M/2.02M) for our experiments. $T^*$ is randomly selected within range $[0, T]$. The experimental results presented are the average of the results obtained by setting up three random seeds. As shown in Tab. 6 BEEF-Net requires only a small number of iterations to converge to the specified slicing time based on the SPA-Loss. For the complex task with random inputs and disturbances in task (II), BEEF-Net can still converge fast and even faster to find the specified cut point compared with task(I). *This simple experiment demonstrates that SPA-Loss can effectively supervise the SNN pulsing at the specified location, which paves the way for experiments on adaptive event slicing in real scenarios.*

## L  STATISTICS OF DYNAMIC SLICING (BEEF) VS. FIXED SLICING

In this section, we compare the statistics results of the resulting events sliced by different slicing methods.

*Symbol Description: the total event stream $E$; the resulting sliced sub-event stream list by BEEF: $E_{beef} = [E_1^b, ..E_{N_1}^b]$; the resulting sliced sub-event stream list by fixed slicing method: $E_{fixtime} = [E_1^f, ..E_{M_1}^f]$.*

In the tracking task, the average duration of each sub-event stream $E_k^b (k \in [1, N_1])$ is 65ms (corresponding to 13 event cells, and the duration of the event stream contained in each event cell is 5ms). The maximum duration of each sub-event stream is 100ms, and the minimum duration is 30ms, while for our comparison of the slicing-by-fixed-time approach, the duration of each sub-event stream $E_j^f (j \in [1, M_1])$ is fixed at 75ms. The following are specific statistics:

| Method | Avg Cell Num | Var Cell Num | Avg Duration | Min Duration | 25th Duration | 75th Duration | Max Duration |
|---|---|---|---|---|---|---|---|
| BEEF | 12.99 | 3.96 | $\sim$ 65ms | 25ms | 50ms | 80ms | 100ms |
| Slice by fixed duration | 15 | 0 | 75ms | // | // | // | // |

Table 7: Statistic results of dynamic slicing method (our BEEF) and fixed slicing method.

## M  STATISTICS OF EVENT DENSITY

We also counted the average density of sub-event streams after fixed slicing and dynamic slicing for comparison.

*Symbol Description: For each sub-event stream $E_k^b$ or $E_j^f$, it contains several event points $e_i = [x_i, y_i, t_i, p_i]$. We define a matrix $C$ to represent the event count, where $C_{xy}$ represents the number of events at coordinates $(x, y)$. Given a threshold $T$, we define the event density to be:*

$$D = \frac{\sum_{x,y} \mathbb{1}\{C_{x,y} \geq T\}}{\sum_{x,y} \mathbb{1}\{C_{x,y} > 0\}}, \tag{26}$$

where $\mathbb{1}$ denotes the indicator function. The threshold $T$ is determined based on the percentile of the event count $C_{xy}$.

**Density Analysis:** We define the density of events as a metric reflecting the amount of event information within sub-event streams. The smaller the fluctuation in the density of each sub-stream (the smaller the variance), the more stable the event information contained. The stability of the event

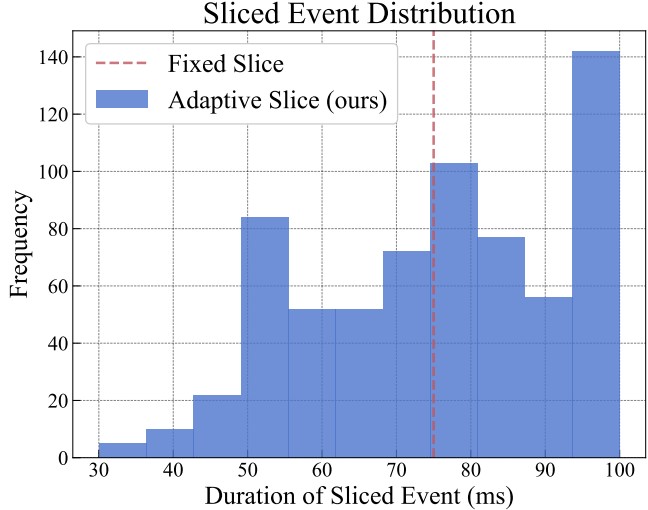

Figure 9: Visualization of sliced event distribution. Our BEEF can return the sliced events with different durations, while the fixed method can only generate sliced events with fixed durations.

density is crucial for scenarios where the distribution of events is not uniform (*e.g.*, scenarios with changing motion speed).

We chose $T = 30\%$ and $90\%$ (lower $T$ to observe overall event activity, including those less frequent events; higher $T$ for focusing on repetitive or frequent events) to validate the effectiveness of BEEF to dynamically slice the event stream. Below are the statistics of event density, where the data (left vs right) on the left are statistics of BEEF and on the right for statistics of fixed slicing. We select 4 classes in the FE108 tracking dataset.

| T=30% | airplane_mul222 | box_hdr | dog | tank_low |
|---|---|---|---|---|
| Mean | 0.9196 vs. 0.9074 | 0.9850 vs. 0.9850 | 0.9208 vs. 0.8984 | 0.9584 vs. 0.9575 |
| Var ↓ | **0.0151** vs. 0.0164 | **0.0038** vs. 0.0039 | **0.0144** vs. 0.0168 | **0.0096** vs. 0.0100 |
| Std ↓ | **0.1231** vs. 0.1283 | **0.0622** vs. 0.0626 | **0.1200** vs. 0.1299 | **0.0984** vs. 0.1003 |

Table 8: Event density comparisons with $T = 30\%$

| T=90% | airplane_mul222 | box_hdr | dog | tank_low |
|---|---|---|---|---|
| Mean | 0.1256 vs. 0.1263 | 0.1338 vs. 0.1405 | 0.1120 vs. 0.1131 | 0.9584 vs. 0.9575 |
| Var ↓ | 0.0004 vs. 0.0004 | **0.0007** vs. 0.0053 | 0.0001 vs. 0.0001 | **0.0096** vs. 0.0100 |
| Std ↓ | 0.0215 vs. 0.0215 | **0.0267** vs. 0.0732 | **0.0073** vs. 0.0078 | **0.0984** vs. 0.1003 |

Table 9: Event density comparisons with $T = 90\%$

The results in Tab. 8 and Tab. 9 show that the event stream after BEEF's dynamic slicing has a **more stable event density** (low variance), which verifies that BEEF has a certain ability to perceive the event information, and ensures that BEEF is robust in different motion scenarios, as shown in Tab. 3.

# N    STATISTICS OF RESULTING SLICING DURATION

In order to further verify the stability and effectiveness of the dynamic slicing method, we explore the results of BEEF by changing the number of event cells in the event recognition task. $N_{cell}$ indicates that an event stream is divided into $N_{cell}$ event cells, and the larger the $N_{cell}$ implies that

the event stream is divided into more fine-grained event cell sequences that are capable of better represent the raw event stream (as mentioned in Section 4.1.).

*Calculation Process: Suppose the whole event stream (duration = $T$) is divided into 15 event cells, if the SNN is trained to sliced the event with 2.42 (average) event cells, which means that the sliced sub-event stream $E_k^b$ contains event data which lasts a duration of $\frac{1}{15} * 2.42 * T = 16.13\%T$.*

| $N_{cell}$ | **15** | **20** | **25** |
|---|---|---|---|
| Avg Cell Num | 2.42 | 3.15 | 4.77 |
| Percentage of Duration | 16.13% | 15.75% | 19.08% |

Table 10: More experiments with latest models in event-based recognition task.

The experimental results show that the percentage of the duration of each sub-event stream to the total event stream duration after the adaptive slicing is relatively stable, *i.e.*, the fineness of the event cell does not affect the event information contained in each sub-event stream after the slicing process, which proves the robustness and effectiveness of the dynamic slicing process of BEEF.

## O   MORE EXPERIMENTS WITH LATEST MODELS

To enhance the credibility and robustness of our results, we have incorporated state-of-the-art models: Swin Transformer (SwinT (Liu et al., 2021)) and Vision Transformer (ViT (Dosovitskiy et al., 2020)), to further validate the efficacy of our BEEF algorithm in event-based recognition tasks:

| **Method** | **Random Slice** | **Fixed Slice** | **Ours** |
|---|---|---|---|
| SwinT (Liu et al., 2021) | 88.19 | 89.93 | **91.67(+1.74%)** |
| ViT (Dosovitskiy et al., 2020) | 87.50 | 85.07 | **88.54(+3.47%)** |

Table 11: Your caption here

*Experiment Settings: We choose SwinT-small and ViT-small for comparisons on DVSGesture dataset. Other settings are consistent with the main experiments.*

BEEF is designed as a **plug-and-play** algorithm for dynamic event stream slicing. It is benchmarked against baselines that employ fixed event stream slicing methods. Our approach is versatile and can be applied in any event recognition or single object tracking framework (including both classical and latest).

## P   MORE EXPERIMENTS ON EXPERT'S ARENA

We put more tracking videos in our supplementary materials.

