# OpenReview forum: "BEEF: Building a BridgE from Event to Frame"
_ICLR.cc/2024/Conference — Submitted to ICLR 2024_

### Official Review · Reviewer_erhw · 2023-10-24

**Soundness:** 3 good
**Presentation:** 4 excellent
**Contribution:** 3 good
**Rating:** 6
**Confidence:** 5

**Summary:**

This paper proposes a novel pre-processing framework (i.e., BEEF) to split continuous event streams into event slices in an adaptive manner. BEEF mainly adopt an energy-efficient SNN to trigger the slicing time. Technically, a new dataset is first split into event slices by SNN, which is robust to high-speed or low-speed scenarios. Then, event slices are used to finetune the ANN to verify the performance in downstream event-based vision tasks. The experiments show that the proposed BEEF achieves SOTA performance in event-based object tracking and event-based object recognition.

**Strengths:**

i) The topic of adaptively splitting event streams using SNN is very interesting and attractive.

ii) The authors sufficient experiments in the main paper and the supplemental material to help reader better understand the main contributions of this work.

iii) The writing is straightforward, clear, and easy to understand.

**Weaknesses:**

i) While fixed windows or a fixed event count may not offer optimal performance for event partitioning pre-processing, they do provide a quick processing option for collaboration with subsequent vision tasks. The authors also adapt the SNN for event stream division, but it's crucial to determine if this process is time-consuming across different platforms (CPU, GPU) and if it's suitable for downstream tasks, particularly those requiring low-latency responses for agile robots. Although the authors give the analysis of processing speed, it should be given the computational analysis in CPU.

ii) The authors have conducted a comparison experiment with a fixed number of times, as shown in Table 3. Nevertheless, it is advisable for the authors to include experiments with a fixed time window. Furthermore, the authors should investigate how various parameters for fixed events or fixed time windows compare to BEEF. Additionally, it would be beneficial for the authors to provide more visual comparison results of event representations.

iii) There are articles exploring adaptive event stream splitting strategies. The author should consider citing some relevant references [1, 2] that utilize hyperparameters for implementation.

[1] EDFLOW: Event driven optical flow camera with keypoint detection and adaptive block matching, IEEE TCSVT 2022.

[2] Asynchronous spatio-temporal memory network for continuous event-based object detection, IEEE TIP 2022.

**Questions:**

See weakness.

---

> ### Author Response · Authors · 2023-11-17
> **Response to erhw (Reply 1)**
>
> **Weakness 1 (Computational Analysis in both GPU and CPU)**
>
> **Q1:** *While fixed windows or a fixed event count may not offer optimal performance for event partitioning pre-processing, they do provide a quick processing option for collaboration with subsequent vision tasks. The authors also adapt the SNN for event stream division, but it's crucial to determine if this process is time-consuming across different platforms (CPU, GPU) and if it's suitable for downstream tasks, particularly those requiring low-latency responses for agile robots. Although the authors give the analysis of processing speed, it should be given the computational analysis in CPU.*
>
> **A1:**
> Thank you for your insightful comments! We have conducted a computational analysis of the BEEF framework on both CPU and GPU platforms:
>
> |  |Slicing Latency |FPS |
> |:------|:----------------|:----------------|
> | SNN on GPU | 0.009s per img | 111 Hz|
> | SNN on CPU | 0.430s per img | 2.3 Hz|
>
>
>  The results indicate that SNN processing of event streams on a GPU offers the advantage of low latency, while there is a noticeable delay when processing on a CPU.
>
> However, it's important to note that SNNs are primarily designed to operate on neuromorphic hardware, where they can leverage their low power consumption and low latency advantages for efficient data processing in scenarios demanding quick responses. There is significant research demonstrating the benefits of SNNs in processing event streams on neuromorphic hardware. For instance, the use of the Loihi hardware for event-based vision tasks [1,2] and the Tianji chip [3] for robotics applications [4] are notable examples.
>
> We will include these details in our revised manuscript to provide a comprehensive understanding of the processing capabilities of SNNs across different hardware platforms. Thanks for your suggestion!
>
> ***Reference:***
>
> [1] Roy A, Nagaraj M, Liyanagedera C M, et al. Live Demonstration: Real-time Event-based Speed Detection using Spiking Neural Networks. CVPRW 2023.
>
> [2] Viale A, Marchisio A, Martina M, et al. Carsnn: An efficient spiking neural network for event-based autonomous cars on the loihi neuromorphic research processor. IJCNN 2021.
>
> [3] Brain-inspired multimodal hybrid neural network for robot place recognition. Science Robotics 2023
>
> [4] Viale A, Marchisio A, Martina M, et al. LaneSNNs: Spiking Neural Networks for Lane Detection on the Loihi Neuromorphic Processor. IROS 2022.

---

> ### Author Response · Authors · 2023-11-17
> **Response to erhw (Reply 2)**
>
> **Weakness 2 (Comparisons on Different Fixed Slicing Methods)**
>
> **Q2:** *The authors have conducted a comparison experiment with a fixed number of times, as shown in Table 3. Nevertheless, it is advisable for the authors to include experiments with a fixed time window. Furthermore, the authors should investigate how various parameters for fixed events or fixed time windows compare to BEEF. Additionally, it would be beneficial for the authors to provide more visual comparison results of event representations.*
>
> **A2:**
> Thank you very much for your valuable suggestions! In the original experiments presented in Table 3, we used the slicing of fixed event count for comparison. We will include more specific details about this in the revised version of our manuscript. To facilitate a more complete comparison between dynamic slicing and traditional fixed slicing methods, we have supplemented our experiments in the object recognition task. These include slicing based on a fixed number of events and a fixed duration of events. We have also compared three different event representation methods, with the results as follows:
>
> | DVSGesture |Event Frame | Event Spike Tensor | Voxel Grid |
> |:------|:----------------|:--------|:----|
> | Fix Duration | 93.75% | 93.75% | 88.54% |
> |Fix Event Count |93.06%|94.79%|88.19%|
> |**BEEF(ours)** |**94.79%**|**95.49%**|**89.24%**|
>
>
> The results show that our dynamic slicing approach, BEEF, outperforms the fixed slicing approach in downstream tasks across different event representations. This highlights the efficacy of BEEF in handling event streams.
>
> Additionally, we plan to include more illustrative figures in the revised version to vividly depict the differences and respective advantages of dynamic and fixed slicing. More visualization results related to the experiments will also be added to the appendix.
>
> Thank you again for your suggestion! Your feedback is instrumental in enhancing the clarity and comprehensiveness of our research.

---

> ### Author Response · Authors · 2023-11-17
> **Response to erhw (Reply 3)**
>
> **Weakness 3 (Relevant References)**
>
> **Q3:** *There are articles exploring adaptive event stream splitting strategies. The author should consider citing some relevant references [1, 2] that utilize hyperparameters for implementation.*
>
> **A3:**
> Thank you very much for your suggestion! We acknowledge the importance of incorporating relevant references, particularly those that explore adaptive event stream splitting strategies. We will make sure to include the references you have mentioned [1,2] in the revised version of our manuscript! Thanks!
>
> ***Reference:***
>
> [1] Liu M, Delbruck T. EDFLOW: Event driven optical flow camera with keypoint detection and adaptive block matching[J]. IEEE TCSVT 2022.
>
> [2] Li J, Li J, Zhu L, et al. Asynchronous spatio-temporal memory network for continuous event-based object detection[J]. IEEE TIP 2022.

---

> > ### Comment · Reviewer_erhw · 2023-11-21
> > **I stuck with the original score.**
> >
> > The author has addressed all my queries through the experimental responses. However, I suggest incorporating these experiments into the supplementary material to make them accessible to a broader audience of authors.

---

> > > ### Author Response · Authors · 2023-11-21
> > > **Responese to erhw (Thanks)**
> > >
> > > Thank you very much for your valuable suggestions and feedback. We are delighted to have addressed all your queries and appreciate your recognition of our work.
> > >
> > > In response to your suggestion, we have submitted a revised manuscript that includes the details of the important experiments. We will submit the final revised version containing all the suggested experiments shortly!
> > >
> > > Thank you once again!

---

### Official Review · Reviewer_689Y · 2023-10-26

**Soundness:** 3 good
**Presentation:** 3 good
**Contribution:** 2 fair
**Rating:** 5
**Confidence:** 3

**Summary:**

The authors propose BEEF, a novel-design event processing framework that can slice the event streams in an adaptive manner. To achieve this, BEEF employs an SNN as the event trigger to dynamically determine the time at which the event stream needs to be split, rather
than requiring hyper-parameter adjustment as in traditional methods.

**Strengths:**

S1: papers dealing with spiking related algorithms should be of interest to the subset of the machine learning community investigating on-the-edge computing algorithms.

S2: the paper is relatively well written

**Weaknesses:**

W1: I am aware that with event and spiking cameras it is quite popular to convert the event/spike streams into a sort of frame based representation. However I have a fundamental objection with this type of an approach (which is shared by quite a few of my colleagues around the world, in private conversations at least) as to why should these fundamentally asynchronous
event streams representations should be converted to a rather synchronous representation, simply to be able to map them into algorithms that were originally developed for synchronous frame like data. I think a more thorough discussion on this is needed in the paper to better motivate the work

W2: clarify better what are the alternative methods to which this is being compared? What exactly is meant by "fixed slice" approaches to which this is being compared? Many approaches for producing frame like representations (such as getting the max or union of all events in a time window) result in the introduction of significant amounts of noise. In contrast morphological operands like erosion and dilation can introduce much better quality frames. To what extent is the good performance of the algorithm attributable simply to noisy frame generation in competing approaches?

W3: unless i missed it, will source code be provided?

**Questions:**

See my questions above. Addressing them would improve the paper's relevance

---

> ### Author Response · Authors · 2023-11-17
> **Response to 689Y (Reply 1)**
>
> **Weakness 1 (Asynchronous Event and Synchronous Representation)**
>
> **Q1:** *I am aware that with event and spiking cameras it is quite popular to convert the event/spike streams into a sort of frame based representation. However I have a fundamental objection with this type of an approach (which is shared by quite a few of my colleagues around the world, in private conversations at least) as to why should these fundamentally asynchronous event streams representations should be converted to a rather synchronous representation, simply to be able to map them into algorithms that were originally developed for synchronous frame like data. I think a more thorough discussion on this is needed in the paper to better motivate the work.*
>
> **A1:**
> Thank you for your insightful comment. First, I wholeheartedly agree with your statement regarding the conversion of asynchronous event data into synchronous formats, which indeed can undermine the inherent advantages of the asynchronous nature. This is not an optimal representation, and we acknowledge this limitation. However, the reasons for converting asynchronous event data into synchronous representations in current practices can be summarized as follows:
>
> 1. Why choose synchronous over asynchronous processing? Given that existing GPU hardware architectures and programming models are designed for highly synchronous and parallel tasks, the conversion of asynchronous event stream data into a synchronous representation becomes a necessity for algorithm simulations performed on GPUs. In our paper, we aim to slice the event stream into very fine event cells to represent the original event data as closely as possible (Sec.4.1), hoping to minimize the errors introduced by synchronous representation.
>
> 2. The synchronous simulations conducted on GPUs in our work do not preclude the possibility of future asynchronous implementations on neuromorphic hardware. Both SNNs and event stream data are inherently asynchronous. However, due to hardware constraints, current software simulations are temporarily unable to achieve true asynchronous processing. If deployed on neuromorphic hardware (e.g., Loihi [1], TrueNorth [2]), the asynchronous processing of event streams by SNNs would be extremely low-energy and low-lantancy[3,4,5]. Therefore, this work lays the theoretical groundwork for future hardware deployment, and efficiently implementing SNN processing of asynchronous event streams in conjunction with ANNs on neuromorphic hardware is one of our future goals.
>
> We hope this explanation addresses your concerns. We are committed to further discussing this topic in our manuscript to provide a comprehensive motivation for our work.
>
>
> ***Reference:***
>
> [1] Davies M, Srinivasa N, Lin T H, et al. Loihi: A neuromorphic manycore processor with on-chip learning. IEEE Micro 2018.
>
> [2] Akopyan F, Sawada J, Cassidy A, et al. Truenorth: Design and tool flow of a 65 mw 1 million neuron programmable neurosynaptic chip. IEEE TCAD 2015.
>
> [3] Roy A, Nagaraj M, Liyanagedera C M, et al. Live Demonstration: Real-time Event-based Speed Detection using Spiking Neural Networks. In CVPRW 2023.
>
> [4] Viale A, Marchisio A, Martina M, et al. Carsnn: An efficient spiking neural network for event-based autonomous cars on the loihi neuromorphic research processor. In IJCNN 2021.
>
> [5] Yu F, Wu Y, Ma S, et al. Brain-inspired multimodal hybrid neural network for robot place recognition. Science Robotics 2023.

---

> ### Author Response · Authors · 2023-11-17
> **Response to 689Y (Reply 2)**
>
> **Weakness 2 (Clarification and Discussion)**
>
> **Q2:** *clarify better what are the alternative methods to which this is being compared? What exactly is meant by "fixed slice" approaches to which this is being compared? Many approaches for producing frame like representations (such as getting the max or union of all events in a time window) result in the introduction of significant amounts of noise. In contrast morphological operands like erosion and dilation can introduce much better quality frames. To what extent is the good performance of the algorithm attributable simply to noisy frame generation in competing approaches?*
>
> **A2:**
>
> *Symbol Description: the total event stream $E$; the resulting sliced sub-event stream list by BEEF: $E_{beef}=[E^b_{1},..E^b_{N_1}]$; the resulting sliced sub-event stream list by fixed slicing method: $E_{fixtime}=[E^f_{1},..E^f_{M_1}]$.*
>
> Thank you for your insightful suggestions! First, let me clarify our dynamic slicing approach, which allows the event stream to be sliced at any timestamp and then converted into frames for downstream tasks. In Table 1, the term "fixed slice" refers to a method that we employ a fixed slicing approach with a constant number of events per sub-event stream to segment the entire event stream into $E_{fixtime}=[E^f_{1},..E^f_{M_1}]$. These sub-event streams are then transformed into different representations using the same event representation method $F$ (we used Event Frame [1]) and fed into downstream tasks (tracking/recognition) for testing. We will revise the terminology to "slicing with fixed event number" in our updated manuscript to make the experimental settings clearer.
>
> We greatly appreciate your mention of the potential use of morphological operands. These methods are more commonly found in event stream denoising algorithms or event representation algorithms [2], and our paper primarily focuses on the event stream slicing process. The techniques you mentioned, like erosion or dilation, could be integrated with our current approach; for example, denoising the original event stream before applying BEEF’s dynamic slicing. We intend to investigate it further in future research. Thank you for this valuable suggestion!
>
> ***Reference:***
>
>
> [1] Maqueda A I, Loquercio A, Gallego G, et al. Event-based vision meets deep learning on steering prediction for self-driving cars. CVPR 2018.
>
> [2] Baldwin R W, Liu R, Almatrafi M, et al. Time-ordered recent event (TORE) volumes for event cameras. TPAMI 2022.

---

> ### Author Response · Authors · 2023-11-17
> **Response to 689Y (Reply 3)**
>
> **Weakness 3 (Code Available)**
>
> **Q3:** *unless i missed it, will source code be provided?*
>
> **A3:**
> Absolutely！We will provide the source code upon acceptance.

---

> ### Comment · Reviewer_689Y · 2023-11-22
>
> I have read the comments by the authors and reviewers. The overall evaluation of the paper seems to be consistent more or less across reviewers. As I indicated in my comments the conversion of asynchronous events to a frame based representation defeats the purpose of event/spiking cameras (in my personal opinion at least). Indicating that this is done because we need to run it on GPUs is not convincing to me, so I keep my ranking unchanged.

---

> ### Author Response · Authors · 2023-11-23
> **Response to 689Y (Clarification)**
>
> Thank you for your response and thank you for taking the time and effort to review our paper. Although BEEF demonstrate superior performance compared with SOTA, but we believe that the flow in the original paper may have caused confusion, for instance, most reviewers confused our method with representation method. We notice that and emphasize our method is not one type of representation methods but compatible with any representation method (including asynchronous representation) in the revised paper. In addition, thanks to the suggestions made by the reviewers, we have added a number of experiments that have significantly increased the reliability of our paper.
>
> Back to your query, to the best of our knowledge, there is no commonly used SNN framework that can directly handle asynchronous inputs; current frameworks can only handle frame-based inputs. However, we believe that the SNN trained with current frameworks can potentially be utilized to process asynchronous input. To demonstrate this, we quickly add one experiment. When the number of slices ($N$) increases, the frame-like input will gradually become more similiar to asynchronous inputs. Therefore, we cut the event stream into up to 1000 slices, each slice has only ~100 events and almost all pixels are 0 or 1. Considering the total pixel number is 180 x 240 = 43200, it is quite sparse. From following table, the same SNN will fire at similar percentages (i.e., the resulting sub-stream always contains similar event point), demonstrating that SNN is capable of perceiving event information. Indeed, it is still not fully asynchronized input, but we hope it would provide some insights here.
>
> | $N$  |          30 |          50 |          100 |          200 |         300 |         400 |         500 |         600 |         700 |         800 |         900 |         1000 |
> |:-------------|:-----------|:-----------|:-----------|:-----------|:-----------|:-----------|:-----------|:-----------|:-----------|:-----------|:-----------|:-----------|
> | Spike Position     |   15 | 25 | 51 | 102  | 153 | 213 | 256 | 324  | 387  | 444 | 504 | 560 |
> | Percentage of Containing Event   | 50.00%    | 50.00%  | 51.00% | 51.00% | 51.00% | 53.25% | 51.20%  | 54.00%  | 55.29% | 55.50% | 56.00% | 56.00% |

---

### Official Review · Reviewer_YMHz · 2023-10-31

**Soundness:** 2 fair
**Presentation:** 3 good
**Contribution:** 2 fair
**Rating:** 5
**Confidence:** 5

**Summary:**

This paper proposes an efficient way for event representation. Specifically, they introduce SNN for adaptive event slicing, which can choose appropriate slicing times considering the events’ temporal feature and downstream task. The authors present several losses to further improve the adaptiveness, and a strategy to let SNN better assist the of ANN in an iterative and cooperative manner.

**Strengths:**

+ The overall writing of this work is clear and easy to follow.
+ The three observations and solutions seem to work well and improve the adaptation for slicing time.
+ Using SNN in event representation is rational considering the similar feature for SNN and event.

**Weaknesses:**

- This paper fails to fully review the topic of this work: event representation. As suggested in [1][2], there are several existing event representation strategies including stacking based on time/event counts, voxel grid, histogram of time surfaces, event spike tensor, and a recent work introduces neural representation [3]. However, this paper only mentions two of them. In addition, the motivation to consider temporal information is similar with event counts integration, which is mentioned by the authors.
- The necessity of a very lightweight SNN is not clear. Since SNN works with ANN cooperatively, SNN has only very limited contribution to the overall computational cost. As implied in Table 2, considering the ANN is the major cost for the process, the contribution and necessity for low energy and fast speed of SNN is reduced.
- The compared methods in the experiment are not sufficient. More event representation/stacking methods should be considered to compare with the proposed methods, including the methods mentioned in [1-3].
- I wonder whether such iterative optimization of SNN and ANN work better than joint optimization, like we regard the whole process as an end-to-end task and optimize the SNN loss and downstream task loss together.
- More details about the experimental settings are required. The proposed methods use adaptive slicing time, how to create GT accordingly? And how to compare with fixed-sliced methods that have different timestamps for event frames?

[1] End-to-End Learning of Representations for Asynchronous Event-Based Data, ICCV 2019
[2] Event-based High Dynamic Range Image and Very High Frame Rate Video Generation using Conditional Generative Adversarial Networks, CVPR 2019
[3] NEST: Neural Event Stack for Event-based Image Enhancement, ECCV 2022

**Questions:**

See the weakness above

---

> ### Author Response · Authors · 2023-11-17
> **Response to YMHz (Reply 1)**
>
> **Weakness 1 (Difference between Event Slicing and Event Representation)**
>
> **Q1:** *This paper fails to fully review the topic of this work: event representation. As suggested in [1][2], there are several existing event representation strategies including stacking based on time/event counts, voxel grid, histogram of time surfaces, event spike tensor, and a recent work introduces neural representation [3]. However, this paper only mentions two of them. In addition, the motivation to consider temporal information is similar with event counts integration, which is mentioned by the authors.*
>
> **A1:**
> Thank you for your recommendations. I understand your concern regarding why our paper does not address a broader range of event representation methods. The reason is that **our focus is on event slicing rather than event representation.** Event stream data conversion into frames/representations involves two steps: Step 1: slicing the event stream into multiple sub-event streams, and Step 2: converting these sub-streams into frames using different event representation methods. There is a significant body of work dedicated to optimizing event representation (Step 2), including the voxel grid, time surface, and other methods you mentioned. However, these do not address the issues arised with fixed slicing (e.g., resulting non-uniform event in scenarios with changing motion speed). Hence, our paper primarily addresses the first step: event slicing.
>
> After the dynamic slicing with BEEF, the events can indeed be transformed into different representational formats using various event representation methods. Although event slicing and representation are different processing steps, either better slicing or representation method benefits the feature extraction with neural network, thus improving performance. Experiments with different slicing methods and different event representation have been supplemented in **Reply 3**.
>
> The references you have mentioned will also be included in the updated version of our manuscript. And the confusing description of motivation in the original article that you pointed out will also be revised shortly!
>
> I hope this explanation addresses your query. Thank you once again!

---

> ### Author Response · Authors · 2023-11-17
> **Response to YMHz (Reply 2)**
>
> **Weakness 2 (Necessity of Using SNN)**
>
> **Q2:** *The necessity of a very lightweight SNN is not clear. Since SNN works with ANN cooperatively, SNN has only very limited contribution to the overall computational cost. As implied in Table 2, considering the ANN is the major cost for the process, the contribution and necessity for low energy and fast speed of SNN is reduced.*
>
> **A2:**
> Thank you for your question. Let me elucidate the necessity of using SNN.
>
> 1.The reason **why we choose SNN as the event slicing trigger** is twofold:
> - Utilizing SNNs on neuromorphic hardware for processing event streams is low-energy and low-latency [1,2].
>
> - Deployed on neuromorphic hardware, SNNs can process event streams asynchronously [3,4,5], conserving energy when there is no data input—a capability that GPUs, operating synchronously, lack.
>
> Due to the aforementioned reasons, there is a considerable amount of research [6,7,8,9,10] employing Spiking Neural Networks (SNNs) for event data. Although these SNNs are simulated on GPU platforms, the models resulting from such simulations could be deployed on neuromorphic hardware [3,4].
>
>
> 2.The rationale behind our aim for **low-energy and fast-speed SNN processing** is:
>
> We design the BEEF as a **plug-and-play** algorithm, intending for the SNN to dynamically slice the event stream without adversely affecting the latency or energy consumption of the main network.
>
> Regarding the comparison of resource consumption between SNNs and ANNs (Table 2), it is crucial to note that our dynamic slicing is performed in real-time, not in an offline manner.
> By employing a low-energy SNN model as a dynamic event stream slicer, we ensure that the speed does not impede the downstream processing rate and also enhances the overall performance of downstream tasks. This is one of the core motivations of our paper, and we hope it addresses your concerns.
>
> ***Reference:***
>
> [1] Davies M, Srinivasa N, Lin T H, et al. Loihi: A neuromorphic manycore processor with on-chip learning. IEEE Micro 2018.
>
> [2] Akopyan F, Sawada J, Cassidy A, et al. Truenorth: Design and tool flow of a 65 mw 1 million neuron programmable neurosynaptic chip. IEEE TCAD 2015.
>
> [3] Roy A, Nagaraj M, Liyanagedera C M, et al. Live Demonstration: Real-time Event-based Speed Detection using Spiking Neural Networks. In CVPRW 2023.
>
> [4] Viale A, Marchisio A, Martina M, et al. Carsnn: An efficient spiking neural network for event-based autonomous cars on the loihi neuromorphic research processor. In IJCNN 2021.
>
> [5] Yu F, Wu Y, Ma S, et al. Brain-inspired multimodal hybrid neural network for robot place recognition. Science Robotics 2023.
>
> [6] Hagenaars J, Paredes-Vallés F, De Croon G. Self-supervised learning of event-based optical flow with spiking neural networks. NeurlPS 2021.
>
> [7] Yao M, Gao H, Zhao G, et al. Temporal-wise attention spiking neural networks for event streams classification. ICCV 2021.
>
> [8] Zhu L, Wang X, Chang Y, et al. Event-based video reconstruction via potential-assisted spiking neural network. CVPR 2022
>
> [9] Kosta A K, Roy K. Adaptive-spikenet: event-based optical flow estimation using spiking neural networks with learnable neuronal dynamics. ICRA 2023.
>
> [10] Hussaini S, Milford M, Fischer T. Spiking neural networks for visual place recognition via weighted neuronal assignments. RAL 2023.

---

> ### Author Response · Authors · 2023-11-17
> **Response to YMHz (Reply 3)**
>
> **Weakness 3 (Comparisions of Different Event Representation)**
>
> **Q3:** *The compared methods in the experiment are not sufficient. More event representation/stacking methods should be considered to compare with the proposed methods, including the methods mentioned in [1-3].*
>
> **A3:**
> Thanks for your suggestion! Event representation refers to the process of event information extraction that is performed after the event stream has been sliced into sub-event stream, and the resulting event representation meets the neural network input requirements. Thus, our dynamic slicing process and event representation can be used at the same time.
>
> To validate the effectiveness of our slicing approach, we assess the downstream task performance using three distinct event representation methods, namely Event Frame [1], Event Spike Tensor (EST) [2], and Voxel Grid [3], on the DVSGesture dataset. We measure these against both fixed (slice by fixed duration and fixed event count) and dynamic slicing approaches to provide a comprehensive analysis:
>
> | DVSGesture |Event Frame | Event Spike Tensor | Voxel Grid |
> |:------|:----------------|:--------|:----|
> | Fix Duration | 93.75% | 93.75% | 88.54% |
> |Fix Event Count |93.06%|94.79%|88.19%|
> |**BEEF(ours)** |**94.79%**|**95.49%**|**89.24%**|
>
>
> The results show that our dynamic slicing approach, BEEF, outperforms the fixed slicing approach in downstream tasks across different event representations. This highlights the efficacy of BEEF in handling event streams.
>
> Thanks for your suggestion! We will cite the event representation methods you mentioned in the revised paper and try to integrate them with BEEF in the future.
>
> ***Reference:***
>
> [1] Maqueda A I, Loquercio A, Gallego G, et al. Event-based vision meets deep learning on steering prediction for self-driving cars. CVPR 2018.
>
> [2] Gehrig D, Loquercio A, Derpanis K G, et al. End-to-end learning of representations for asynchronous event-based data. ICCV 2019.
>
> [3] Zhu A Z, Yuan L, Chaney K, et al. Unsupervised event-based learning of optical flow, depth, and egomotion. CVPR 2019.

---

> ### Author Response · Authors · 2023-11-17
> **Response to YMHz (Reply 4)**
>
> **Weakness 4 (End-to-end Optimization)**
>
> **Q4:** *I wonder whether such iterative optimization of SNN and ANN work better than joint optimization, like we regard the whole process as an end-to-end task and optimize the SNN loss and downstream task loss together.*
>
> **A4:**
> Thanks for your constructive suggestion! We have explored your idea of training the SNN and ANN from scratch and optimizing them together. However, as shown in the table below, the results of ResNet18 have a similar performance to the original results, while ResNet34 even has a performance degradation:
>
> | DVSGesture | Random Slice| Fixed Slice | BEEF | BEEF (optimize both from scratch) |
> |:------|:----------------|:--------|:----|:----|
> | ResNet18 | 93.06% | 93.40% | 93.49% |93.75% |
> |ResNet34 |95.14%|93.40%|96.18%| 92.36%|
>
> Thus, we believe that further in-depth exploration is needed to fully realize the concept you've mentioned and the training strategies need to be improved.
>
> Thank you for interesting comments!

---

> ### Author Response · Authors · 2023-11-17
> **Response to YMHz (Reply 5)**
>
> **Weakness 5 (Experiment Details)**
>
> **Q5:** *More details about the experimental settings are required. The proposed methods use adaptive slicing time, how to create GT accordingly? And how to compare with fixed-sliced methods that have different timestamps for event frames?*
>
> **A5:**
> Thank you for your question. Below are more detailed explanations of our experimental settings and the methodology for generating Ground Truth (GT). We will include these details in the appendix of the revised version:
>
> **Experimental settings：**
>
> 1. *Network Structure*:
> We used a Spiking Neural Network architecture comprising {16C3-IF-AP2-32C3-IF-AP2-64C3-IF-AP2-LNIF-LN-IF}, where IF denotes the use of integrated-fire neurons. (Appendix E)
>
> 2. *Training Setup*:
> We adopted the SGD optimizer with an initial learning rate of 1e-4, complemented by a cosine learning rate scheduler. SNN models were trained for 50 epochs with a batch size of 32. (Appendix E)
>
> 3. *Data Processing Setup:*
> For single object tracking tasks, we utilized the FE108 dataset, while for object recognition, we used the DVS-gesture and N-caltech101 datasets. Each dataset was divided into training, testing, and validation sets. The validation set was used to infer the ANN model to provide feedback for supervising SNN training. All results in the table represent the ANN's performance on the test set.
>
> 4. *GT Setting:*
> Since the SNN might choose to spike and segment the event stream at any position, the GT at any timestamp was obtained through linear interpolation from the GT provided by the original dataset.
>
> **Details of Fixed-sliced method:**
>
> *Symbol Description: the total event stream $E$; the resulting sliced sub-event stream list by BEEF: $E_{beef}=[E^b_{1},..E^b_{N_1}]$; the resulting sliced sub-event stream list by fixed slicing method: $E_{fixtime}=[E^f_{1},..E^f_{M_1}]$.*
>
> Suppose an event stream (duration = $T$) is sliced into $N'$ slices ($E_{beef}=[E^b_{1},..E^b_{N_1}]$) after dynamic slicing. Then for a fair comparison, the number of sub-event stream generated by the fixed slicing method should also be $N'$ (i.e., $len(E_{fixtime})=N'$), where the duration of each sub-event stream in $E_{fixtime}$ is $\frac{T}{N'}$. Since it was mentioned earlier that we know the GT of each timestamp, we can likewise obtain the GT corresponding to each sub-event stream after fixed slicing. Then we can compare the results of the fixed slicing method with our dynamic slicing method.

---

### Official Review · Reviewer_q3dC · 2023-10-31

**Soundness:** 3 good
**Presentation:** 2 fair
**Contribution:** 3 good
**Rating:** 3
**Confidence:** 4

**Summary:**

This paper is about the slicing step in the conversion from events to binned representations that can yield frames for classical image processing.
The goal of this paper is to make the event-slicing step adaptive instead of fixed over time as it is now in the majority of the approaches that use slicing/binning/bucketing where events are assigned to slices with slices being constant time length or containing equal numbers of events.

The way it works is that events are fed to a spiking neural network with Leaky Integrate and Fire neurons. The SNN fires more sparsely than the original events.
A new slice is created containing all events between the timings of two output spikes.

To control the desired time offset of the slice a membrane potential loss is introduced. Authors give a formal proof for the sufficient conditions.
Moreover, a linear assuming loss resolves the dependence between neighboring membrane potentials.

Experiments are conducted on object tracking and gesture/object recognition with impressive results.

**Strengths:**

1. The dynamic slicing of events using the output spikes of an SNN.

2. The connection between slicing and downstream task expressed in the additional two loss terms determining the hyperparameters of the SNN.

3. The theoretical treatment of the sufficient condition of firing at a desired time (given in the appendix).

**Weaknesses:**

1. Frame-like inputs to transformers or CNNs where frames have been derived from events may be sensitive to slicing. We need a toy experiment to study this hypothesis with a smaller network and different slicing techniques.

2. The exposition is really hard to follow. As stated directly after eq. 4, the slicing is done by grouping together events whose timestamps are between two output spikes of the SNN. Here, an experiment is needed on the statistics of this slicing and why such an approach makes sense.

3. 4.3.1 has to be elaborated. While the math derivations are sound, it is not clear to the reader why the starting point of the derivations is the desire for $S_{out}$ to spike at $n^{*}$. I tried to understand it also through the observations in 4.3.2 but could not.

4. The beginner's arena was meant to explain the above but is incomprehensible. What does it mean ``to slice at a specified time step $T^{*}'' ?

5. It is not clear what purpose the energy computations of the SNN serve when the task will be solved with ultra consuming GPUs.

6. The experimental comparison should be with approaches that are asynchronous end to end like HOTS or HATS or Cannici'19, Perot'20 etc. or approaches like the Event Transformer.

7. Table 3: It is not discussed why the transformer tracker performs almost the same or better without BEEF. Why does BEEF not add anything significant when an attention mechanism is used?

8. The feedback strategy is learnt during training. I understand that in this sense it is adaptive to the task rather than during inference to the event stream when the hyperparameters will be fixed.

9. It is unclear whether events are treated differently according to their polarity.

10. There is some problem with the definition of ${\cal D}$ because $n_q$ is not defined anywhere but mentioned ``where $n_q$ denotes the time of the last spike''.

11. It would be worth listing the latency from event to GPU output for the particular architectures on tracking and recognition. This is much more critical here than the power consumption of the CNN.

Summary: The authors need to explain the slicing method more clearly (possible misreadings are listed above). My main concern is the lack of any experimental analysis or motivation for the particular quite elaborate slicing method. There is no motivation to use an SNN since the slicing is only a minimal energy and latency fraction of a pipeline that uses transformers or regression.
There is no comparison with architectures that use other event representations like time surfaces.

**Questions:**

Weaknesses are numbered and should be considered as questions.

---

> ### Author Response · Authors · 2023-11-17
> **Response to q3dC (Reply 1)**
>
> **Weakness 1 (Slicing Sensitivity)**
>
> **Q1:** *Frame-like inputs to transformers or CNNs where frames have been derived from events may be sensitive to slicing. We need a toy experiment to study this hypothesis with a smaller network and different slicing techniques.*
>
> **A1:**
> Thank you very much for your constructive suggestion! In response, we have conducted total 60 experiments with **different models** to investigate the impact of **different slicing techniques** and **different number of slices** on the performance in downstream tasks, thereby affirming the hypothesis that event streams are sensitive to slicing.
>
> In our experiment, we employed two fixed slicing methods: (1). *Slicing with a fixed number of events* and (2). *Slicing with a fixed duration*. $N$ denotes the number of resulting event slices. Experimental results are detailed as follows:
>
> | NCaltech101   | $N$  |          2 |          4 |          6 |          8 |         10 |         12 |         14 |         16 |         18 |         20 |         22 |         24 |         26 |         28 |         30 |     Mean|     Var|
> |:---------------|:-------------|:-----------|:-----------|:-----------|:-----------|:-----------|:-----------|:-----------|:-----------|:-----------|:-----------|:-----------|:-----------|:-----------|:-----------|:-----------|:-----------|:-----------|
> | ResNet18       | Fixed Count     |   70.96 | 75.26 | 75.39 | 75.30  | 76.09 | 73.95 | 74.09 | 73.80  | 76.40  | 75.39 | 75.45 | 73.60  | 71.94 | 71.01 | 71.17 | **73.98**|**3.33**|
> | ResNet18       | Fixed Time     | 62.90  | 72.64 | 76.38 | 74.48 | 74.91 | 73.70  | 74.30  | 74.69 | 76.95 | 74.75 | 74.46 | 74.42 | 71.61 | 71.52 | 69.69 |**73.16**|**10.80**|
> | ResNet34       | Fixed Count| 72.19 | 75.55 | 76.98 | 78.22 | 77.14 | 77.40  | 76.78 | 76.90  | 78.14 | 77.06 | 76.91 | 74.85 | 74.76 | 76.91 | 73.07 |**76.19**|**2.90**|
> | ResNet34       | Fixed Time| 65.42 | 75.92 | 78.29 | 78.20  | 78.48 | 76.22 | 77.76 | 76.57 | 75.94 | 76.80  | 76.61 | 75.91 | 75.11 | 74.76 | 74.19 |**75.74**|**9.15**|
>
> The results indicate significant fluctuations (large variance) in downstream performance based on the slicing method and the number of slices used. We believe this addition effectively demonstrates the sensitivity of event streams to fixed slicing techniques, confirming the need for our motivation to propose dynamic slicing of event streams.
> Additionally, the accuracy achieved using the dynamic slicing method (82.54% by ResNet34) surpasses that of any fixed slicing approach (with the highest being 78.48%), further substantiating the efficacy of the dynamic method in our study.
>
> Thanks for your suggestion!

---

> > ### Author Response · Authors · 2023-11-17
> > **Response to q3dC (Reply 2.2)**
> >
> > **Continued from Reply 2.1**
> >
> > ### **3. Statistics of Resulting Slicing Duration.**
> >
> > In order to further verify the stability and effectiveness of the dynamic slicing method, we explore the results of BEEF by changing the number of event cells in the event recognition task. $N_{cell}$ indicates that an event stream is divided into $N_{cell}$ event cells, and the larger the $N_{cell}$ implies that the event stream is divided into more fine-grained event cell sequences that are capable of better represent the raw event stream (as mentioned in Section 4.1.).
> >
> > | $N_{cell}$ |15 | 20 | 25 |
> > |:------|:----------------|:--------|:----|
> > | Avg Cell Num | 2.42 | 3.15 | 4.77 |
> > |Percentage of Duration |16.13%|15.75%|19.08%|
> >
> > *Calculation Process: Suppose the whole event stream (duration = $T$) is divided into 15 event cells, if the SNN is trained to sliced the event with 2.42 (average) event cells, which means that the sliced sub-event stream $E^b_k$ contains event data which lasts a duration of $\frac{1}{15}\times2.42T = 16.13$%$T$.*
> >
> > The experimental results show that the percentage of the duration of each sub-event stream to the total event stream duration after the adaptive slicing is relatively stable, i.e., the fineness of the event cell does not affect the event information contained in each sub-event stream after the slicing process, which proves the robustness and effectiveness of the dynamic slicing process of BEEF.
> >
> > ### **Summary**
> > We have taken care to conduct an in-depth statistical analysis of the dynamic slicing process employed by our BEEF algorithm. This analysis, alongside the demonstrated enhancements in downstream task performance as shown in Tables 1 and 3 of our manuscript, reinforces the validity of our approach. We are confident that the slicing mechanism of BEEF, which groups events between two output spikes of the SNN, is both statistically sound and practically effective.

---

> ### Author Response · Authors · 2023-11-17
> **Response to q3dC (Reply 2.1)**
>
> **Weakness 2 (Statistics of Slicing Method)**
>
> **Q2:** *The exposition is really hard to follow. As stated directly after eq. 4, the slicing is done by grouping together events whose timestamps are between two output spikes of the SNN. Here, an experiment is needed on the statistics of this slicing and why such an approach makes sense.*
>
> **A2:**
> To demonstrate the effectiveness of our proposed dynamic slicing method, we provide the following statistical results:
>
> ### **1. Statistics of dynamic slicing (BEEF) vs. fixed slicing.**
> *Symbol Description: the total event stream $E$; the resulting sliced sub-event stream list by BEEF: $E_{beef}=[E^b_{1},..E^b_{N_1}]$; the resulting sliced sub-event stream list by fixed slicing method: $E_{fixtime}=[E^f_{1},..E^f_{M_1}]$.*
>
> In the tracking task, the average duration of each sub-event stream $E^b_{k}(k\in[1,N_1])$ is 65ms (corresponding to 13 event cells, and the duration of the event stream contained in each event cell is 5ms). The maximum duration of each sub-event stream is 100ms, and the minimum duration is 30ms, while for our comparison of the slicing-by-fixed-time approach, the duration of each sub-event stream $E^f_{j}(j\in[1,M_1])$ is fixed at 75ms. The following are specific statistics:
>
> |Method| Avg Cell Num| Var Cell Num | Avg Duration| Min Duration | 25th Duration| 75th Duration| Max Duration|
> |:--------|:--------------------------------------|:---------------------------------------|:------------------------------|:--------|:-----|:-----|:----|
> | BEEF                                  |  12.99                          | 3.96    | ~65ms                                  |25ms    | 50ms   | 80ms  | 100ms |
> | Slice by fixed duration               |  15                             | 0       | 75ms                                  |//   | //   | //  | //  |
>
> We will put the visualization of the statistic results in the appendix of the revised version.
>
> ### **2. Statistics of event density.**
>
> We also counted the average density of sub-event streams after fixed slicing and dynamic slicing for comparison.
>
> *Symbol Description: For each sub-event stream $E^b_{k}$ or $E^f_{j}$, it contains several event points $e_i=[x_i,y_i,t_i,p_i]$. We define a matrix $C$ to represent the event count, where $C_{xy}$ represents the number of events at coordinates $(x,y)$. Given a threshold $T$, we define the event density to be: $D = \frac{\sum_{x,y}\mathbb{1}\{C_{x,y}\geq T\}}{\sum_{x,y}\mathbb{1}\{C_{x,y}> 0\}}$. The threshold $T$ is determined based on the percentile of the event count $C_{xy}$.*
>
> **Density Analysis**: We define the density of events as a metric reflecting the amount of event information within sub-event streams. The smaller the fluctuation in the density of each sub-stream (the smaller the variance), the more stable the event information contained. The stability of the event density is crucial for scenarios where the distribution of events is not uniform (e.g., scenarios with changing motion speed).
>
>
> We chose $T = 30$% and $90$% (lower $T$ to observe overall event activity, including those less frequent events; higher $T$ for focusing on repetitive or frequent events) to validate the effectiveness of BEEF to dynamically slice the event stream. Below are the statistics of event density, where the data (left vs right) on the left are statistics of BEEF and on the right for statistics of fixed slicing. We select 4 classes in the FE108 tracking dataset.
>
> | T=30% | airplane_mul222 | box_hdr | dog | tank_low |
> |:------|:----------------|:--------|:----|:---------|
> | Mean  | 0.9196 vs 0.9074 | 0.9850 vs 0.9850 | 0.9208 vs 0.8984 | 0.9584 vs 0.9575 |
> | Var $\downarrow$| **0.0151** vs 0.0164 | **0.0038** vs 0.0039 | **0.0144** vs 0.0168 | **0.0096** vs 0.0100 |
> | Std $\downarrow$ | **0.1231** vs 0.1283 | **0.0622** vs 0.0626 | **0.1200** vs 0.1299 | **0.0984** vs 0.1003 |
>
> | T=90% |airplane_mul222 | box_hdr | dog | tank_low|
> |:------|:----------------|:--------|:----|:---------|
> | Mean | 0.1256 vs 0.1263 | 0.1338 vs 0.1405 | 0.1120 vs 0.1131 | 0.9584 vs 0.9575 |
> | Var $\downarrow$| 0.0004 vs 0.0004 | **0.0007** vs 0.0053 | 0.0001 vs 0.0001 | **0.0096** vs 0.0100 |
> | Std $\downarrow$| 0.0215 vs 0.0215 | **0.0267** vs 0.0732 | **0.0073** vs 0.0078 | **0.0984** vs 0.1003 |
>
> The results show that the event stream after BEEF's dynamic slicing has a more stable event density (low variance), which verifies that BEEF has a certain ability to perceive the event information, and ensures that BEEF is robust in different motion scenarios, as shown in Table 3.

---

> ### Author Response · Authors · 2023-11-17
> **Response to q3dC (Reply 3)**
>
> **Weakness 3 (Explaination of our method)**
>
> **Q3:** *4.3.1 has to be elaborated. While the math derivations are sound, it is not clear to the reader why the starting point of the derivations is the desire for $S_{out}$ to spike at $n^*$. I tried to understand it also through the observations in 4.3.2 but could not.*
>
> **A3:**
> Thank you for your query which provides us with an opportunity to clarify our methodology.
>
> Contrary to fixed slicing methods (such as predetermined time intervals or event counts), our approach dynamically determines event stream slicing based on spike occurrences in the SNN, detailed in Figure 1.
>
> In order to supervise the SNN to slice the event stream at the optimal position, we feed the events sliced at the SNN-determined position as well as the events sliced at its neighboring positions into the downstream model.
> The downstream model then returns the loss (e.g., classification loss, tracking loss), where the position corresponding to the minimum loss indicates the optimal slicing point (Eq.10). Thus, we obtain a position label $n^*$, which in turn guides the SNN to make the best slicing strategy through supervised learning.
>
> In summary, when ANN feedback indicates that slicing at the $n^*$ position can enhance model performance, the SNN discriminator is directed to spike at the $n^*$. Section 4.3 of our paper delves into formulating the SPA-CE loss function, which aids the SNN to spike at this specified $n^*$ position.
>
> We hope this response clarifies our approach and methodology. We are committed to improving the manuscript in its revised version for better understanding. Should there be any points needing further clarification, please feel free to reach out.

---

> ### Author Response · Authors · 2023-11-17
> **Response to q3dC (Reply 4)**
>
> **Weakness 4 (Explaination of $T$*)**
>
> **Q4:** *The beginner's arena was meant to explain the above but is incomprehensible. What does it mean ``to slice at a specified time step $T$ * '' ?*
>
>
> **A4:**
> Following up on the explanation of Reply 3, consider the downstream ANN's feedback indicates that the SNN should spike at a specific moment $n^*$ to slice the event stream. In this case, our goal is to supervise the SNN to ensure that it spikes precisely at $n^*$. The purpose of "The beginner's arena" is to demonstrate that our proposed loss function SPA-CE can effectively guide the SNN to pulse at $n^*$, in contrast to common loss functions like Cross-Entropy (CE) or Mean Squared Error (MSE), which could not achieve this.
>
> The term $T^*$ used in the text may have caused some confusion. In future revisions of our manuscript, we will standardize this notation to $n^*$ to avoid any ambiguity. We hope this clarification helps and apologize for any confusion caused. Thank you for the reminder.

---

> ### Author Response · Authors · 2023-11-17
> **Response to q3dC (Reply 5)**
>
> **Weakness 5 (Energy Computation)**
>
> **Q5:** *It is not clear what purpose the energy computations of the SNN serve when the task will be solved with ultra consuming GPUs.*
>
> **A5:**
> Thank you for your comment. Indeed, you are correct that the implementation of Spiking Neural Networks (SNNs) on GPUs does not currently confer a significant energy advantage. This is largely due to the limitations inherent in current SNN simulation platforms (such as spikingjelly[9] and tonic[10]), which are primarily GPU-based.
>
> However, it is important to note that SNNs demonstrate significant energy efficiency when operated on neuromorphic hardware, such as Loihi [7] and TrueNorth [8]. This advantage is well-established and extensively discussed in recent literature [1,2]. The energy calculations in our paper [3] are intended to provide an estimation of the potential energy efficiency of SNNs with future advancements in hardware and algorithms. In addition, comparing the theoretical energy consumption of SNNs with that of traditional Artificial Neural Networks (ANNs) is commonly-adopted in this field [4,5,6], aiding in the understanding of the energy dynamics of these systems.
>
> ***Reference:***
>
> [1] Yin B, Corradi F, Bohté S M. Accurate online training of dynamical spiking neural networks through Forward Propagation Through Time. Nature Machine Intelligence 2023.
>
> [2] Schuman C D, Kulkarni S R, Parsa M, et al. Opportunities for neuromorphic computing algorithms and applications. Nature Computational Science 2022.
>
> [3] Yao M, Zhao G, Zhang H, et al. Attention spiking neural networks. TPAMI 2023.
>
> [4] Kim S, Park S, Na B, et al. Spiking-yolo: spiking neural network for energy-efficient object detection. AAAI 2020.
>
> [5] Zhou Z, Zhu Y, He C, et al. Spikformer: When spiking neural network meets transformer. ICLR 2023.
>
> [6] Wang Z, Fang Y, Cao J, et al. Masked Spiking Transformer. ICCV 2023.
>
> [7] Davies M, Srinivasa N, Lin T H, et al. Loihi: A neuromorphic manycore processor with on-chip learning. IEEE Micro 2018.
>
> [8] Akopyan F, Sawada J, Cassidy A, et al. Truenorth: Design and tool flow of a 65 mw 1 million neuron programmable neurosynaptic chip. IEEE TCAD 2015.
>
> [9] Fang W, Chen Y, Ding J, et al. SpikingJelly: An open-source machine learning infrastructure platform for spike-based intelligence. Science Advances 2023.
>
> [10] Eshraghian J K, Ward M, Neftci E O, et al. Training spiking neural networks using lessons from deep learning. Proceedings of the IEEE 2023.

---

> ### Author Response · Authors · 2023-11-17
> **Response to q3dC (Reply 6)**
>
> **Weakness 6 (Comparisions of Different Event Representation)**
>
> **Q6:** *The experimental comparison should be with approaches that are asynchronous end to end like HOTS or HATS or Cannici'19, Perot'20 etc. or approaches like the Event Transformer.*
>
> **A6:**
> Thank you for your suggestion! The above methods you mentioned are all about event representation methods. It is worth noting that **our work focuses on the slicing of the event stream rather than focusing on event representation.** Event representation refers to the process of event information extraction that is performed after the event stream has been sliced into sub-event stream, and the resulting event representation meets the neural network input requirements. Thus, our dynamic slicing process and event representation can be used at the same time, either better slicing or representation method benefits the feature extraction with neural network, thus improving performance.
>
> To validate the effectiveness of our slicing approach, we supplement the event-based recognition task below. We compare the downstream performance of three different event representation methods (including Event Frame [1], Event Spike Tensor (EST [2]) and Voxel Grid [3]) on the DVSGesture dataset under fixed slicing and dynamic slicing:
>
>
> | DVSGesture |Event Frame | Event Spike Tensor | Voxel Grid |
> |:------|:----------------|:--------|:----|
> | Fix Duration | 93.75% | 93.75% | 88.54% |
> |Fix Event Count |93.06%|94.79%|88.19%|
> |**BEEF(ours)** |**94.79%**|**95.49%**|**89.24%**|
>
>
> The results demonstrate that across different event representation methods, our dynamic slicing approach, BEEF, outperforms fixed slicing methods in downstream tasks. This underscores the efficacy of BEEF in handling event streams.
>
> Thanks for your suggestion! We will cite the event representation methods you mentioned in the revised paper and try to integrate them with BEEF in the future.
>
> ***Reference:***
>
> [1] Maqueda A I, Loquercio A, Gallego G, et al. Event-based vision meets deep learning on steering prediction for self-driving cars. CVPR 2018.
>
> [2] Gehrig D, Loquercio A, Derpanis K G, et al. End-to-end learning of representations for asynchronous event-based data. ICCV 2019.
>
> [3] Zhu A Z, Yuan L, Chaney K, et al. Unsupervised event-based learning of optical flow, depth, and egomotion. CVPR 2019.

---

> ### Author Response · Authors · 2023-11-17
> **Response to q3dC (Reply 7)**
>
> **Weakness 7 (Improvement over baseline)**
>
> **Q7:** *Table 3: It is not discussed why the transformer tracker performs almost the same or better without BEEF. Why does BEEF not add anything significant when an attention mechanism is used?*
>
> **A7:**
> Thanks! In fact, for TransT, there are only very few metrics where BEEF does not perform better than baselines, and here's a specific analysis of the amount of improvement in TransT results:
>
> || HDR     |      |      |      | LL      |      |      |      | FWB     |      |      |      | FNB     |      |      |      | ALL     |      |      |      |
> |:--------|:--------|:----|:----|:----|:--------|:----|:----|:----|:--------|:----|:----|:----|:--------|:----|:----|:----|:--------|:----|:----|:----|
> || RSR     | OP.50| OP.75| RPR  | RSR     | OP.50| OP.75| RPR  | RSR     | OP.50| OP.75| RPR  | RSR     | OP.50| OP.75| RPR  | RSR     | OP.50| OP.75| RPR  |
> | TransT (1)   | 55.9 | 71.0   | 24.6 | **84.5**    | 66.8 | 88.9 | 34.3 | 96.5    | 74.1 | **98.6** | 54   | **99.9**    | 55.8 | 69.2 | 24.9 | **85.4**    | 59.6 | 76.4 | 29   | 88.8 |
> | TransT+fixed slice (2)| 51.4 | 67.8 | 11.1 | 81.2    | 63.2 | 80.2 | 28.3 | 89.3    | 41.5 | 28   | 2.5  | 57.7    | 50.6 | 57.9 | 12.7 | 78.9    | 51   | 59   | 12   | 78.8 |
> | TransT+BEEF (3) | **57.7** | **75.2** | **28.1** | 82.6    | **70.7** | **93.6** | **42.7** | **99.0**      | **74.9** | 97.7 | **61.1** | 98.6    | **58.7** | **75.6** | **29.6** | 84.6    | **62.4** | **81.2** | **34.1** | **88.9** |
> | *Improvement* (3)-(1)     | 1.8  | 4.2  | 3.5  | -1.9     | 3.9  | 4.7  | 8.4  | 2.5      | 0.8  | -0.9 | 7.1  | -1.3     | 2.9  | 6.4  | 4.7  | -0.8     | 2.8  | 4.8  | 5.1  | 0.1  |
> |  *Improvement* (3)-(2)        | 6.3  | 7.4  | 17   | 1.4      | 7.5  | 13.4 | 14.4 | 9.7      | 33.4 | 69.7 | 58.6 | 40.9     | 8.1  | 17.7 | 16.9 | 5.7      | 11.4 | 22.2 | 22.1 | 10.1 |
>
> The results demonstrates that 16 out of 20 metrics outperform baselines, and the total average improvement is 6.61%. The bolding value in Table 3 of our original submitted paper indicates a global optimum, which may cause confusion. We'll change this in our revised version shortly, thanks!

---

> ### Author Response · Authors · 2023-11-17
> **Response to q3dC (Reply 8)**
>
> **Weakness 8 (Learnable Feedback Strategy)**
>
> **Q8:** *The feedback strategy is learnt during training. I understand that in this sense it is adaptive to the task rather than during inference to the event stream when the hyperparameters will be fixed.*
>
> **A8:**
> That's right! The feedback strategy is indeed learned during the training process. Therefore, during inference, we utilize the trained SNN to slice the event stream instead of using a fixed set of hyperparameters for this task. The event frames sliced by the SNN are then directly fed into the downstream ANN model for testing.

---

> ### Author Response · Authors · 2023-11-17
> **Response to q3dC (Reply 9)**
>
> **Weakness 9 (Polarity Process)**
>
> **Q9:** *It is unclear whether events are treated differently according to their polarity.*
>
>
> **A9:**
> We apologize for not talking about the handling of event polarity in the main text. Our approach retains the polarity information of the events. The information for each polarity is accumulated separately. Additionally, we have compared our method with other event representation techniques, which are presented in Reply 6. We will ensure to clearly articulate these experimental details in the subsequent version of our manuscript.

---

> ### Author Response · Authors · 2023-11-17
> **Response to q3dC (Reply 10)**
>
> **Weakness 10 (Missing Definition)**
>
> **Q10:** *There is some problem with the definition of $D$ because $n_q$ is not defined anywhere but mentioned ``where
> $n_q$ denotes the time of the last spike''.*
>
>
> **A10:**
> My apologies for the confusion caused by the writing error in Section 4.2. The phrase "where $n_q$ denotes the time of the last spike" should indeed be removed as it was not previously defined. Thank you for bringing this to our attention. We will correct this error shortly!

---

> ### Author Response · Authors · 2023-11-17
> **Response to q3dC (Reply 11)**
>
> **Weakness 11 (Latency Computation)**
>
> **Q11:** *It would be worth listing the latency from event to GPU output for the particular architectures on tracking and recognition. This is much more critical here than the power consumption of the CNN.*
>
>
> **A11:**
> I agree. It's of great significance to consider the latency of in the processing of events, especially in practical applications. As documented in Table 2 of our manuscript, the Frames Per Second (FPS) rate achieved by utilizing the SNN to slice the event stream is 111Hz, which corresponds to processing once every 0.009 seconds. This rate is significantly lower than the latency experienced by the downstream ANN models during tracking tasks, which have an FPS rate of 39. Consequently, our BEEF model can facilitate real-time dynamic event slicing.
>
>
> |  |Latency | FPS |
> |:------|:----------------|:--------|
> | SNN | 0.009s per img | 111 Hz|
> |ANN |0.025s per img|39 Hz|

---

> ### Author Response · Authors · 2023-11-17
> **Response to q3dC (Reply 12.1 Summary 1)**
>
> Thanks again for all your suggestions and questions! Your valuable suggestions greatly help us to improve the content and quality of our articles! Please allow me to revisit and clarify the motivation behind our dynamic event stream slicing algorithm and how we've verified its effectiveness.
>
> ### **1. Motivation for Proposing a Dynamic Event Stream Slicing Algorithm**
>
> Let's start by clarifying the process of event-to-frame conversion, which is mainly divided into two steps: **Step 1. Slice the raw event stream into multiple sub-event stream** and **Step 2. convert these sub-event streams into frames using various event representation methods.** While many works has focused on optimizing event representation (Step 2) to extract better event information, including time surface and EST, they do not address the issues arised with fixed slicing (e.g., resulting non-uniform event in scenarios with changing motion speed). Despite event slicing being a small part of the overall pipeline, it is a critical point. This is because the event stream is very sensitive to slicing, and the model performance fluctuates very much for different slicing methods, as proved by extensive experiments in Reply 1.
>
> To better address this issue, we introduced the dynamic slicing framework BEEF. Meanwhile, BEEF is guided by downstream task feedback to ensure that the new sub-streams could enhance downstream task performance.
>
>
>
>
> ### **2. Motivation for Using SNN as a Slicing Trigger**
>
> The reason why we choose SNN as the event slicing trigger is twofold:
>
> - Utilizing SNNs on neuromorphic hardware for processing event streams is low-energy and low-latency [1,2].
>
> - Deployed on neuromorphic hardware, SNNs can process event streams asynchronously [3,4,5], conserving energy when there is no data input—a capability that GPUs, operating synchronously, lack.
>
> Due to the aforementioned reasons, there is a considerable amount of research [6,7,8,9,10] employing Spiking Neural Networks (SNNs) for event data. Although these SNNs are simulated on GPU platforms, the models resulting from such simulations could be deployed on neuromorphic hardware [3,4].
>
> ### **3. Contribution for Using SNN as a Slicing Trigger**
> We propose a new cooperative paradigm where SNN acts as an efficient, low-energy data processor to assist the ANN in improving downstream
> performance. This is a brand-new SNN-ANN cooperation way, paving the way for future event-related implementation on neuromorphic chips.
>
> ### **4. Comparison with Other Event Representations**
> Although our article focuses on dynamic event stream slicing, not event representation, we appreciate your suggestion that verifying the effectiveness of dynamic slicing BEEF through different event representations is necessary. As I mentioned earlier, event slicing and event representation are not in conflict, and a fusion of the two may have potentially better enhancements. Experiments with different slicing methods and different event representation have been supplemented in Reply 6.

---

> ### Author Response · Authors · 2023-11-17
> **Response to q3dC (Reply 12.2 Summary 2)**
>
> **Continued from Reply 12.1.**
>
> ### **Summary**
> This article focuses on dynamic event slicing methods and also proposes a cooperative ANN-SNN paradigm for future deployment on hardware.
> We are deeply grateful for your advice! We are making every effort to resolve any confusion and will improve sections that could cause misunderstandings in the updated version, such as why we guide the SNN to pulse at $n^*$ and the difference between event slicing and event representation. If anything is still unclear, we will promptly revise and respond! Thanks!
>
>
> ***Reference:***
>
> [1] Davies M, Srinivasa N, Lin T H, et al. Loihi: A neuromorphic manycore processor with on-chip learning. IEEE Micro 2018.
>
> [2] Akopyan F, Sawada J, Cassidy A, et al. Truenorth: Design and tool flow of a 65 mw 1 million neuron programmable neurosynaptic chip. IEEE TCAD 2015.
>
> [3] Roy A, Nagaraj M, Liyanagedera C M, et al. Live Demonstration: Real-time Event-based Speed Detection using Spiking Neural Networks. In CVPRW 2023.
>
> [4] Viale A, Marchisio A, Martina M, et al. Carsnn: An efficient spiking neural network for event-based autonomous cars on the loihi neuromorphic research processor. In IJCNN 2021.
>
> [5] Yu F, Wu Y, Ma S, et al. Brain-inspired multimodal hybrid neural network for robot place recognition. Science Robotics 2023.
>
> [6] Hagenaars J, Paredes-Vallés F, De Croon G. Self-supervised learning of event-based optical flow with spiking neural networks. NeurlPS 2021.
>
> [7] Yao M, Gao H, Zhao G, et al. Temporal-wise attention spiking neural networks for event streams classification. ICCV 2021.
>
> [8] Zhu L, Wang X, Chang Y, et al. Event-based video reconstruction via potential-assisted spiking neural network. CVPR 2022
>
> [9] Kosta A K, Roy K. Adaptive-spikenet: event-based optical flow estimation using spiking neural networks with learnable neuronal dynamics. ICRA 2023.
>
> [10] Hussaini S, Milford M, Fischer T. Spiking neural networks for visual place recognition via weighted neuronal assignments. RAL 2023.

---

### Official Review · Reviewer_48Pk · 2023-11-01

**Soundness:** 2 fair
**Presentation:** 2 fair
**Contribution:** 2 fair
**Rating:** 5
**Confidence:** 3

**Summary:**

The paper studies to learn event splits by using SNN. The triggered spikes from SNN are treated as signals for splitting event streams and constructing event frames. The proposed architecture is evaluated with object recognition and single object tracking datasets.

**Strengths:**

* The motivation of the paper is well demonstrated. Fixed event stream fixed slicing methods potentially fail to generalize in different motion scenarios.
* How the paper finds optimal spike time, $n_{s}$, is interesting.
* The paper shows relative improvements over different baseline methods when using their proposed BEEF framework.

**Weaknesses:**

* The paper claims a fixed event split method fails to generalize. However,  event cell $C[N]$ is a discrete 2D representation generated from a fixed event split, and is used as the input for SNN.
* BEEF can be used in ANN-based 3D CNN/Transformer seamlessly. Event cameras and SNN are all bio-inspired but do not necessarily imply that SNN is a good fit to event data.

**Questions:**

* Why not experiment with the latest event recognition/single object tracking framework? The latest methods in Tab. 1 and Tab. 3 were published in 2021?

---

> ### Author Response · Authors · 2023-11-17
> **Response to 48Pk (Reply 1)**
>
> **Weakness 1 (Event Cell)**
>
> **Q1:** *The paper claims a fixed event split method fails to generalize. However, event cell $C[N]$ is a discrete 2D representation and used as the input for SNN is generated from a fixed event split.*
>
> **A1:**
> Thank you for your comment! You are correct in noting that our event cells adopt a 2D format. Ideally, our goal is for the Spiking Neural Network (SNN) to process raw event inputs asynchronously. However, due to the constraints that CNN-based deep learning frameworks can only accept 2D inputs, we also adopted 2D format, which is a commonly-used method for processing event [1,2,3].
>
> In our work, we have endeavored to slice the entire event stream into very fine event cells along the timeline as much as possible, since we want to find the optimal and accurate event slices.
> These event cells contain very little information, and our SNN is able to process the event cell and supervised to slice the event stream adaptively.
>
> In the future, we expect to further our research by deploying SNNs on neuromorphic hardware. This would enable the asynchronous processing of event streams using BEEF, potentially overcoming current constraints and enhancing the efficiency of our approach.
>
> ***Reference:***
>
> [1] Zhang J, Yang X, Fu Y, et al. Object tracking by jointly exploiting frame and event domain. ICCV 2021.
>
> [2] Baldwin R W, Liu R, Almatrafi M, et al. Time-ordered recent event (TORE) volumes for event cameras. TPAMI 2022.
>
> [3] Maqueda A I, Loquercio A, Gallego G, et al. Event-based vision meets deep learning on steering prediction for self-driving cars. CVPR 2018.

---

> ### Author Response · Authors · 2023-11-17
> **Response to 48Pk (Reply 2)**
>
> **Weakness 2 (Match between SNN and Event)**
>
> **Q2:** *BEEF can be used in ANN-based 3D CNN/Transformer seamlessly. Event cameras and SNN are all bio-inspired but do not necessarily imply that SNN is a good fit to event data.*
>
> **A2:**
> Thank you for your comment!
> In the original article, we mentioned that ``both SNN and event stream are brain-like inspired, which motivates us to use SNN to process events'', and there is indeed a problem with the logic of this statement. I appreciate you reminding us of this point. Therefore, here we have reorganized the motivation for using SNN to process events:
>
> - Utilizing SNNs on neuromorphic hardware for processing event streams is low-energy and low-latency [1,2].
>
> - Deployed on neuromorphic hardware, SNNs can process event streams asynchronously [3,4,5], conserving energy when there is no data input—a capability that GPUs, operating synchronously, lack.
>
> Due to the aforementioned reasons, there is a considerable amount of research [6,7,8,9,10] employing Spiking Neural Networks (SNNs) for event data. Although these SNNs are simulated on GPU platforms, the models resulting from such simulations could be deployed on neuromorphic hardware [3,4].
>
> We hope this explanation satisfactorily addresses your concerns and illustrates the rationale behind our methodology.
>
> ***Reference:***
>
> [1] Davies M, Srinivasa N, Lin T H, et al. Loihi: A neuromorphic manycore processor with on-chip learning. IEEE Micro 2018.
>
> [2] Akopyan F, Sawada J, Cassidy A, et al. Truenorth: Design and tool flow of a 65 mw 1 million neuron programmable neurosynaptic chip. IEEE TCAD 2015.
>
> [3] Roy A, Nagaraj M, Liyanagedera C M, et al. Live Demonstration: Real-time Event-based Speed Detection using Spiking Neural Networks. In CVPRW 2023.
>
> [4] Viale A, Marchisio A, Martina M, et al. Carsnn: An efficient spiking neural network for event-based autonomous cars on the loihi neuromorphic research processor. In IJCNN 2021.
>
> [5] Yu F, Wu Y, Ma S, et al. Brain-inspired multimodal hybrid neural network for robot place recognition. Science Robotics 2023.
>
> [6] Hagenaars J, Paredes-Vallés F, De Croon G. Self-supervised learning of event-based optical flow with spiking neural networks. NeurlPS 2021.
>
> [7] Yao M, Gao H, Zhao G, et al. Temporal-wise attention spiking neural networks for event streams classification. ICCV 2021.
>
> [8] Zhu L, Wang X, Chang Y, et al. Event-based video reconstruction via potential-assisted spiking neural network. CVPR 2022
>
> [9] Kosta A K, Roy K. Adaptive-spikenet: event-based optical flow estimation using spiking neural networks with learnable neuronal dynamics. ICRA 2023.
>
> [10] Hussaini S, Milford M, Fischer T. Spiking neural networks for visual place recognition via weighted neuronal assignments. RAL 2023.

---

> ### Author Response · Authors · 2023-11-17
> **Response to 48Pk (Reply 3)**
>
> **Weakness 3 (Latest Methods)**
>
> **Q3:** *Why not experiment with the latest event recognition/single object tracking framework? The latest methods in Tab. 1 and Tab. 3 were published in 2021?*
>
> **A3:**
> To enhance the credibility and robustness of our results, we have incorporated state-of-the-art models: Swin Transformer (SwinT) and Vision Transformer (ViT), to further validate the efficacy of our BEEF algorithm in event-based recognition tasks:
>
>
> | **Method** | Random Slice | Fixed Slice | **Ours** |
> |:-------|:-------|:-------|:------|
> | SwinT | 88.19 | 89.93 | **91.67(+1.74%)**|
> | ViT | 87.50 | 85.07 |**88.54(+3.47%)**|
>
> *Experiment Settings: We choose SwinT-small and ViT-small for comparisons on DVSGesture dataset. Other settings are consistent with the main experiments.*
>
> BEEF is designed as a **plug-and-play** algorithm for dynamic event stream slicing. It is benchmarked against baselines that employ fixed event stream slicing methods. Our approach is versatile and can be applied in any event recognition or single object tracking framework (including both classical and latest).
>
> We appreciate your suggestion and hope that this response adequately addresses your query.

---

> ### Comment · Reviewer_48Pk · 2023-11-23
> **Event Cell**
>
> Thank you for the clarification. I will maintain my rating. Alternative approaches such as representing event data as a graph could potentially address the conflict.

---

> > ### Author Response · Authors · 2023-11-23
> > **Response to 48Pk (Clarification)**
> >
> > Thanks for your reply. We appreciate your time and effort for reviewing our paper. We understand that it would be easy to confuse our method with representation method. Because current common slicing methods are based on fixed durations or event counts, which can lead to redundancy or lack of sufficient information, our approach is therefore to adaptively slice the event stream into sub-streams. Afterwards, the sub-streams can be converted into various event representations, such as frame, graph, voxel, etc. We choose frame as the representation in the ANN part only because it is mostly used in literature. Our adaptive slicing method will find the sub-steams with just enough information to avoid redundancy, and could be compatible with any representation method to improve the performance. Hope above explanation resolve your confusion between slicing method and representation, and more details can be found in *##Response to q3dC (Reply 12.1 Summary 1)* or *##Response to YMHz (Reply 1)*. Moreover, in our paper, we clearly mention that event slicing and event representation are different steps (in paragraph 2 & 3 in Section 1) to avoid the confusion between these two concepts.

---

### Author Response · Authors · 2023-11-17
**Overall Response to Reviewers**

We sincerely thank the reviewers for their thoughtful comments and feedback. We appreciate that all reviewers agreed that the idea of using spiking neural network (SNN) for dynamic event slicing is interesting and the  theoretical analysis is sound. Below, we address the primary concerns raised by the reviewers:

**1.Motivations of Using SNN for Dynamic Event Slicing.**

The choice of Spiking Neural Networks (SNNs) for dynamic event slicing is driven by their ability to process asynchronously in a low-energy and low-latency manner, when deployed on neuromorphic hardware. Based on these reasons, substantial works apply SNNs on event data. Although these SNNs are simulated on GPU platforms, the outcomes of these simulations lay the groundwork for their potential deployment on neuromorphic hardware.


**2.Distinguishing Between Event Slicing and Event Representation.**

Event-to-frame conversion typically entails two key steps: (1) slicing the event stream into multiple sub-event streams (eg. slicing with fixed duration or event counts), and (2) transforming these sub-streams into frame-like formats using various event representation methods (e.g. voxel grid, time surface and event spike tensor). Our paper primarily addresses the first step — event slicing — aiming to alleviate the issues arised with fixed slicing (e.g., resulting non-uniform event in scenarios with changing motion speed). Although they are different processing steps, either better slicing or representation method benefits the feature extraction with neural network, thus improving performance.


**3. Additional Experiments and Statistical Analysis.**

We sincerely appreciate your effort in providing constructive feedback! In response, we have incorporated additional experiments to further demonstrate that our dynamic event slicing method is solid and effective.

- Reviewer 48Pk suggested the need for including the latest experimental models for comparison. In response, we supplemented the classification task with several of the latest models and verified that our approach also delivers improvements.
- Reviwer q3dC argues that there should be experiments to prove that event streams are sensitive to sliced. To address this, we conducted more than 60 ablation experiments to prove the sensitivity of event slicing does exists.
- Reviewer q3dC pointed out the importance of more comprehensive statistical analysis to demonstrate the effectiveness of our approach. Therefore, we compiled statistical results of dynamic slicing in three perspectives, including event density, duration and position, reinforcing the soundness and efficacy of our method.
- Reviewer q3dC,YMHz and erhw recommended the comparison of our method with various event representation techniques. Thus, we conducted 9 experiments with 3 event representation methods and 3 different slicing methods, substantiating the superiority of our proposed method.
- Reviewer q3dC, YMHz and erhw provided suggestions for SNN processing latency and GPU deployment. We have thus provided latency statistics for SNN on GPU and detailed the distinctions between SNN deployment on GPU and neuromorphic hardware.

Thanks for your time and effort in reviewing our paper, your suggestions have greatly helped to enhance the article!

---

> ### Author Response · Authors · 2023-11-20
> **Overall Response to Reviewers (Reply2)**
>
> We have uploaded an revised manuscript. To facilitate the reviewer's understanding and identification of the significant modifications made to the paper, all crucial revisions have been highlighted in **red** within the revised manuscript.

---

> ### Author Response · Authors · 2023-11-22
> **Overall Response to Reviewers (Reply3)**
>
> I sincerely appreciate the time and effort everyone has invested in providing valuable suggestions for our work. Your comments have significantly enhanced the quality of our work. We hope our responses have satisfactorily addressed all your queries. If there are any further questions or clarifications needed, please feel free to reach out, and we will respond promptly.
>
> As a gentle reminder, we are nearing the end of the review period. Your timely feedback would be greatly appreciated to ensure the smooth progression of our submission.
>
> Thank you once again for your dedication and support！

---

### Comment · Area_Chair_PhBQ · 2023-11-19
**Please engage in reviewer-author discussion**

Dear reviewers,

The paper got diverging scores. The authors have provided their response to the comments. Could you look through their response and  other reviews and engage into the discussion with authors? See if their response changes your assessment of the submission?

Thanks!
AC

---

### Meta-Review · Area_Chair_PhBQ · 2023-12-07

**Metareview:**

This work presents a method for processing event streams through learning-based dynamic slicing which is achieved by Spiking Neural Networks (SNNs). SNN is employed as an event trigger to determine the slicing time based on the spike generation such that events stream could be converted into frames in an adaptive manner. A spiking position-aware loss and feedback-update training strategy is proposed to promote the learning of such SNN. The proposed event slicing method is integrated with several event representation methods and is validated on event-based object tracking and recognition tasks.
Strengths:
This work is well motivated since event representation with fixed slicing may fail to deal with various motions.
potentially fail to generalize in different motion scenarios.
The learning-based dynamic slicing method brings performance gain when integrated with several event representation methods on two tasks.

Weaknesses:
Several reviewers have concerns about the presentation of this work which cause much difficulty in understanding the proposed method. There is no comparison between the proposed adaptive slicing with synchronous event representation and asyncronous event representation like HOTS and HATS.
Since the proposed method first slice the entire event stream into very fine event cells, there could be other options to deal with such event cells for further grouping other than SNN. Authors do not give mcuh explanation on the necessity of using SNN.

**Justification For Why Not Higher Score:**

Several reviewers pointed out that the exposition of the paper was hard to follow. The explanation on relation between the proposed dynamic event slicing and asynchronous event representation is not clear. Authors also do not give mcuh explanation on the necessity of using SNN in the dynamic slicing.

**Justification For Why Not Lower Score:**

It is recommended with Reject

---

### Decision · Program_Chairs · 2024-01-16

Reject